# Particulate-bound alkyl nitrate pollution and formation mechanisms in Beijing, China

Jiyuan Yang[1*], Guoyang Lei[1*], Jinfeng Zhu[1], Yutong Wu[1], Chang Liu[1], Kai Hu[1], Junsong Bao[2], Zitong Zhang[1], Weili Lin[1] and Jun Jin[1,3].

[1]College of Life and Environmental Sciences, Minzu University of China, Beijing 100081, China

[2]State Key Laboratory of Water Environment Simulation, School of Environment, Beijing Normal University, Beijing, 100875, China

[3]Beijing Engineering Research Center of Food Environment and Public Health, Minzu University of China, Beijing 100081, China

*These authors contributed equally to this work

*Correspondence to:* Jun Jin (junjin3799@126.com)

**Abstract**

Fine particulate matter ($PM_{2.5}$) samples were collected between November 2020 and October 2021 at the Minzu University of China in Beijing and the *n*-alkyl nitrate concentrations in the $PM_{2.5}$ samples were determined to investigate *n*-alkyl nitrate pollution and formation mechanisms. $C_9$–$C_{16}$ *n*-alkyl nitrate standards were synthesized and the *n*-alkyl nitrate concentrations in $PM_{2.5}$ were determined by gas chromatography triple quadrupole mass spectrometry. Temporal trends in and correlations between particulate-bound *n*-alkyl nitrate, ozone, $PM_{2.5}$, and nitrogen dioxide concentrations were investigated to assess the relationships between particulate-bound *n*-alkyl nitrate concentrations and gas-phase homogeneous reactions in the photochemical process and speculate the particulate-bound *n*-alkyl nitrates formation mechanisms. The *n*-alkyl nitrate concentrations in the $PM_{2.5}$ samples were 9.67–2730 pg/m³, and the mean was 578 pg/m³. The *n*-alkyl nitrate homologue group concentrations increased as the carbon chain length increased, i.e., long-chain *n*-alkyl nitrates contributed more than short-chain *n*-alkyl nitrates to the total *n*-alkyl nitrate concentrations in $PM_{2.5}$. The *n*-alkyl nitrate concentrations clearly varied seasonally and diurnally, the concentrations decreasing in the order winter > spring > autumn > summer and the mean concentrations being higher at night than in the day. The particulate-bound *n*-alkyl nitrate and ozone concentrations significantly negatively correlated despite gas-phase alkyl nitrate and ozone concentrations previously being found to positively correlate. This indicated that long-chain alkyl nitrates may not be produced during gas-phase homogeneous reactions. The particulate-bound *n*-alkyl nitrate concentrations followed the same trends as and significantly positively correlated with the $PM_{2.5}$ and nitrogen dioxide concentrations. Nitrogen dioxide is an important contributor of nitrates in particulate matter. This indicated that particulate-bound *n*-alkyl nitrates may form through non-homogeneous reactions between alkanes and nitrates on particulate matter surfaces. As secondary pollutants, particulate-bound alkyl nitrates are important components of $PM_{2.5}$ during haze events and strongly affect haze pollution and atmospheric visibility.

## 1 Introduction

Air pollution problems in China are complex but have been alleviated by adjusting the energy structure

and controlling pollutant emissions (Li et al., 2017). However, air pollution (caused by frequent sandstorms in spring, photochemical pollution with ozone and secondary particles forming in summer and autumn, and serious haze pollution caused by emissions caused by heating buildings in winter) remains a problem in urban areas in North China (Bai et al., 2018). Air quality in China will therefore continue to pose serious challenges for some time.

Photochemical smog and haze are important types of air pollution that affect ambient air quality. Interactions between photochemical pollution and particulate pollution have become the main foci of air pollution research (Ma et al., 2012). Nitrogen oxide (NOx) emissions have increased by >50% in the last 30 years (Liu et al., 2013) and NOx concentrations in the atmosphere continue to increase as the number of vehicles increases (Richter et al., 2005; Mijling et al., 2013). Before the NOx reach saturation, more oxidation potentially occurs in the atmosphere as NOx concentrations increase, meanwhile the contributions of anthropogenic emissions to volatile organic compound (VOC) concentrations in the atmosphere are also increasing (Liu et al., 2020). Challenges caused by synergistic photochemical smog and haze pollution are affecting urban areas in which background NOx concentrations are high and large amounts of anthropogenic VOCs are emitted. Future improvements in ambient air quality require both photochemical and particulate pollution to be controlled. Organic nitrates (ONs) formed in the atmosphere from the precursors NOx and VOCs are important atmospheric pollutants, they reflect both photochemical processes of ozone production and SOA formation.

As a kind of semi-permanent reservoir species, ONs are important participants in the atmospheric nitrogen cycle, which involves various atmospheric sources and sinks of nitrogen oxides. The formation of ONs consumes nitrogen oxides and atmospheric oxidants, thus becomes an important sink for atmospheric nitrogen oxides (Perring et al., 2010) and affects the atmospheric lifetimes of free radicals, the ozone concentration, and photochemical reactions (Calvert et al., 1987). In addition, ONs may release nitrogen dioxide and produce strong oxidants such as hydroxyl radicals by photolysis, affecting the balance of nitrogen oxides in regional NOx cycles (Barnes et al., 1993; Chen et al., 1998) and contribute to atmospheric oxidation capacity (Gen et al., 2022), respectively. Semi-volatile ONs are important kind of sources and component of secondary organic aerosols (SOAs) and contribute to fine particulate matter (PM$_{2.5}$) (Rollins et al., 2012). As important secondary air pollutants, ONs affect the oxidation in the atmosphere and the formation of haze (Browne et al., 2012), controlling particulate-bound ONs may therefore be key to controlling both PM$_{2.5}$ and ozone in the atmosphere.

Particulate-bound ONs are some of the main components of particulate matter in China, particularly during pollution events, and strongly affect human health, air quality, and the climate at the regional scale (Zhai et al., 2023). The formation of particulate-bound ONs associated with non-homogeneous reactions (Zhen et al., 2022; Li et al., 2022), especially at night was highly correlated with nitrogen oxide levels. During strong air pollution events, SOAs can contribute up to 30%-77% of PM$_{2.5}$, with particulate organic nitrates accounting for 5%-40% of the organic matter (Rollins et al., 2012; Xu et al., 2015; Sun et al., 2012). ONs have been found to be bound to atmospheric particles in various size ranges (Garnes et al., 2002), indicating that ONs are widely present in atmospheric particulate matter. The strong correlation between ONs and SOAs and the diurnal trend of ONs particle size distribution indicate the key role of particulate-bound ONs (Yu et al., 2019). Recent studies of particulate-bound ONs have mainly been focused on biogenic ONs formed from precursors such as the olefins pinene

(Shen et al., 2021; Rindelaub et al., 2015), limonene (Spittler et al., 2006), monoterpene (Barnes et al., 1990), and isoprene (Rollins et al., 2009; Perring et al., 2009; Vasquez et al., 2020; Wu et al., 2020) emitted from plants. Less attention has been paid to particulate-bound ONs that are related to emissions of anthropogenic pollutants.

Alkyl nitrates are common ONs. Alkanes, as the precursors of alkyl nitrates, have been found to be the most abundant species and contributing 54.1-64.7% of the total VOC concentration (Li et al., 2020), and they were the main components of anthropogenic VOCs that are widely present in the atmosphere (Wei et al., 2018; Kang et al., 2018). It has been found that short-chain ($C_1$–$C_5$) alkyl nitrates are secondary products of photochemical reactions between alkanes and OH· radicals in the gas phase (Jordan et al., 2008; Lim et al., 2009; Perring et al., 2013; Sun et al., 2018), so are associated with photochemical pollution (Simpson et al., 2006; Wang et al., 2013; Ling et al., 2016). The vapour pressure decreases as the carbon chain length increases, so long-chain alkyl nitrates tend to enter the particle phase through gas–particle partitioning and can participate in particulate matter formation and contribute to haze pollution (Lim et al., 2005; Yee et al., 2012). Alkyl nitrates in particulate matter have not received attention in the past, few studies of particulate-bound alkyl nitrates have been performed. Yang et al. developed a gas chromatography triple quadrupole mass spectrometry (GC-MS/MS) method for determining $n$-alkyl nitrate concentrations and detecting $n$-alkyl nitrates in real $PM_{2.5}$ samples (Yang et al., 2019). This indicated that $n$-alkyl nitrates can be present in airborne particulate matter in urban areas. Particulate-bound alkyl nitrates as a kind of secondary pollutants affected by anthropogenic emissions have an important influence on the oxidation of the atmospheric environment and the formation of regional haze pollution (Browne et al., 2012), so it is important to improve our understanding of particulate-bound alkyl nitrate pollution characteristics, temporal variations, and formation mechanisms.

In this study, we determined the concentrations of $C_9$–$C_{16}$ $n$-alkyl nitrates in $PM_{2.5}$ samples collected in Beijing in 2020 and 2021. The aim was to investigate $n$-alkyl nitrate pollution and assess temporal variations in $n$-alkyl nitrate compositions and concentrations. We also assessed the similarities in temporal trends and correlations between the particulate-bound $n$-alkyl nitrate, ozone, $PM_{2.5}$, and nitrogen dioxide ($NO_2$) concentrations to investigate the mechanisms involved in the formation of particulate-bound alkyl nitrates. The study was performed to improve our understanding of alkyl nitrates in $PM_{2.5}$ and improve our ability to control haze pollution.

## 2 Materials and methods

### 2.1 Sampling period and location

Beijing is a typical densely populated large city in China. The heavy traffic in Beijing means that large amounts of exhaust gases are emitted by motor vehicles, and this causes serious haze pollution. Large amounts of anthropogenic $n$-alkanes are emitted to the atmosphere and act as precursors for particulate-bound alkyl nitrates (Kang et al., 2018; Cui et al., 2021). Haidian District is a relatively prosperous area in Beijing. Haidian District is a busy area with high traffic flows and heavy traffic, making it suitable for studying anthropogenic alkyl nitrates in particulate matter. This study was performed at the Minzu University of China (116.19° E, 39.57° N) in Haidian District. $PM_{2.5}$ samples were collected on the roof (about 20 m above the ground) of the College of Pharmacy at the Minzu

University of China. Samples were collected in November and December 2021 and March, April, July, September, and October 2022. Separate day and night samples were collected for one week (23rd to 29th) in each of these months. Each day-time sample was collected from 07:00 to 20:00 and each night-time sample was collected from 20:30 to 06:30 according to the morning and evening rush hours in Beijing, which tend to be 7-9 am and 5-8 pm, respectively.

**2.2 Sample collection and pretreatment**

Each $PM_{2.5}$ sample was collected at a flow rate of 16.7 L/min using a TH-16A low flow sampler (Wuhan Tianhong, Wuhan, China) containing a Whatman QMA quartz fibre filter (Ø 47 mm; GE Healthcare Bio-Sciences, Pittsburgh, PA, USA). Before use, the quartz fibre filters were baked at 550 °C for 5 h to remove organic matter. Each sample was wrapped in aluminium foil and stored at −20 °C.

The *n*-alkyl nitrates in a $PM_{2.5}$ sample were extracted using an ultrasonic extraction method that was described in detail in previous publications (Yang et al., 2019; Yang et al., 2023). The filter was cut into pieces and extracted with 15.0 mL of dichloromethane for 15 min with ultrasonication. The extraction step was repeated five times and the extracts were combined and evaporated to 2.0 mL using a rotary evaporator. The extract was then transferred to a 15 mL centrifuge tube and centrifuged at 3000 rpm for 5 min. The supernatant was then evaporated almost to dryness under a stream of high-purity nitrogen and transferred into 100 μL toluene for instrumental analysis. The sample pretreatment processes were performed with light excluded to prevent photolysis of nitrates.

**2.3 Synthesis and examination of standards**

Standards of *n*-alkyl nitrates could not be purchased, so we synthesized $C_9$–$C_{16}$ *n*-alkyl nitrate standards by performing substitution reactions involving treating brominated *n*-alkanes with silver nitrate using a previously published method (Luxenhofer et al., 1994; Luxenhofer et al., 1996; Yang et al., 2019).

The standards were examined and analyzed by GC-MS/MS, and detected by full scan detection. According to the total ion flow diagrams and mass spectra obtained by GC-MS/MS, only one compound showed a high instrumental response in the total ion flow diagrams, indicating the high purity of synthesized standards. The characteristic ions of *n*-alkyl nitrates, $[CH_2ONO_2]^+$ ion (m/z 76.07) and $[NO_2]^+$ ion (m/z 46.07) appeared in the mass spectra and have high relative abundance, indicating the synthesized standards are *n*-alkyl nitrates.

**2.4 Instrumental analysis**

The *n*-alkyl nitrates ($C_9$–$C_{16}$) were qualitatively and quantitatively analysed using a Trace 1310 gas chromatograph and TSQ 8000 Evo triple quadrupole mass spectrometer (Thermo Fisher Scientific, Waltham, MA, USA). Separation was achieved using a J&W Scientific DB-5M column (30 m long, 0.25 mm inner diameter, 0.1 μm film thickness; Agilent Technologies, Santa Clara, CA, USA). The injection volume was 1.0 μL and splitless injection mode was used. The carrier gas was high-purity helium and the flow rate was 1.0 mL/min. The oven temperature program started at 60 °C, which was held for 3 min, then increased at 10 °C/min to 280 °C, which was held for 3 min. The triple quadrupole mass spectrometer was used in electron impact ionization mode. The ion source temperature was

280 °C and the transmission line temperature was 290 °C. The mass spectrometer was used in selected ion detection mode and $n$-alkyl nitrates were detected by monitoring the characteristic $[NO_2]^+$ ion (m/z 46.07) and $[CH_2ONO_2]^+$ ion (m/z 76.07), which were used as the confirmation and quantitation ions. The GC-MS/MS data were processed and the $n$-alkyl nitrates were quantified using TraceFinder 2.0 software (Thermo Fisher Scientific).

**2.5 Quantitative analysis**

The $n$-alkyl nitrates were quantified using an external standards method. We used the synthesized $C_9$–$C_{16}$ $n$-alkyl nitrates to prepare standard solutions at concentrations of 1000, 100, 50, 20, and 10 ng/mL. A calibration curve was plotted for each analyte with the concentrations of the standards on the $x$-axis and the GC-MS/MS instrument responses on the $y$-axis. The linear ranges of the standard curves for the $C_9$–$C_{16}$ $n$-alkyl nitrate homologues were 10–1000 ng/mL, and the correlation coefficients were all >0.998. The $n$-alkyl nitrate concentrations in the $PM_{2.5}$ sample extracts were quantified using the calibration curves.

**2.6 Quality assurance and control**

Measured and spiked blanks were extracted with each batch of samples. The $n$-alkyl nitrate concentrations found in the blank samples were subtracted from the $n$-alkyl nitrate concentrations found in the samples. The detection and quantification limits of the GC-MS/MS instrument were defined as the concentrations giving signal-to-noise ratios of 3 and 10, respectively. The instrument detection limits for the $n$-alkyl nitrates were 1.0–10.0 pg and the method quantification limits were 0.1–1.0 pg/m³.

The recoveries of the $n$-alkyl nitrates in the $PM_{2.5}$ samples were determined by performing spike recovery experiments, and the recovery was defined as the ratio between the measured and spiked concentrations. Three parallel spiked blank samples were analysed, and 20 µL of a standard solution containing each $C_9$–$C_{16}$ $n$-alkyl nitrate at a concentration of 100 ng/mL was added to each. The spiked blanks were then treated and analysed using the method described above. The $n$-alkyl nitrate concentrations in the spiked blank samples were determined by GC-MS/MS and the recoveries were calculated. The $n$-alkyl nitrate recoveries were 62.6%–95.3% and the relative standard deviation was 2.65%.

**2.7 Data analysis**

The $PM_{2.5}$, ozone, and $NO_2$ concentrations were obtained from the China Meteorological Administration (www.cma.gov.cn/, last access: 31 October 2021). The particulate-bound $n$-alkyl nitrate concentration data were statistically analysed using SPSS 26.0 software (IBM, Armonk, NY, USA). Correlations between concentrations of different species were identified by performing Pearson correlation and Spearman correlation tests (two-tailed), and differences between the concentrations in different samples were assessed by performing independent sample t-tests, paired sample t-tests, and one-way ANOVAs.

**3 Results and discussion**
**3.1 Particulate-bound *n*-alkyl nitrates pollution**
**3.1.1 Concentrations and compositions**
The $C_9$–$C_{16}$ *n*-alkyl nitrates were detected in the $PM_{2.5}$ samples collected during day and night in all of
the seasons, and the concentrations are shown in Figure 1. The concentration ranges, mean
concentrations, and detection rates for the different homologues are shown in Table 1.
The $C_9$ and $C_{10}$ *n*-alkyl nitrate detection rates were <50%, the $C_{11}$ *n*-alkyl nitrate detection rate was
~70%, and the $C_{12}$–$C_{16}$ *n*-alkyl nitrate detection rates were ~90%. The particulate-bound *n*-alkyl nitrate
detection rates generally increased as the carbon chain length increased. These results indicated that
particulate-bound *n*-alkyl nitrates are widely present in airborne particulate matter in Beijing. For
*n*-alkyl nitrates with a single functional group, relatively long chain *n*-alkyl nitrates ($C_{12}$-$C_{16}$) are more
abundant than relatively short chain *n*-alkyl nitrates ($C_9$-$C_{11}$).
The total $C_9$–$C_{16}$ *n*-alkyl nitrate concentrations were 9.67–2730 pg/m³, and the mean was 578 pg/m³. As
shown in Table 1, the particulate-bound *n*-alkyl nitrate homologue concentration range and mean
increased as the carbon chain length increased. The $C_{16}$ *n*-alkyl nitrate homologue had the largest
concentration range, and the mean concentration was significantly higher than the mean concentrations
of the other homologues ($p<0.01$). The $C_{12}$–$C_{16}$ *n*-alkyl nitrate concentrations were significantly higher
than the $C_9$–$C_{11}$ *n*-alkyl nitrate concentrations ($p<0.01$), i.e., the long-chain *n*-alkyl nitrate
concentrations were higher than the short-chain *n*-alkyl nitrate concentrations in the $PM_{2.5}$ samples.
The particulate-bound *n*-alkyl nitrate homologue group compositions in the day and night in the
different seasons during the sampling period are shown in Figures 2 and 3. It can be seen that the $C_{12}$,
$C_{14}$, $C_{15}$, and $C_{16}$ *n*-alkyl nitrate homologues made relatively high contributions to the total *n*-alkyl
nitrate concentrations and that *n*-alkyl nitrates with longer carbon chains ($C_{12}$–$C_{16}$) generally
contributed more than *n*-alkyl nitrates with shorter carbon chains ($C_9$–$C_{11}$) to the total *n*-alkyl nitrate
concentrations during the sampling period.
The long-chain *n*-alkyl nitrate concentrations and contributions to the total *n*-alkyl nitrate
concentrations in $PM_{2.5}$ may have been high because of high concentrations of precursor *n*-alkanes in
the atmosphere and the abilities of *n*-alkyl nitrates to form on airborne particles. *n*-Alkane volatility
decreases as the carbon chain length increases, and long-chain *n*-alkanes are more abundant than
short-chain *n*-alkanes in airborne particulate matter. Our previous study found that the concentration of
precursor *n*-alkanes in $PM_{2.5}$ in Beijing ranged from 4.51 ng/m³ to 153 ng/m³ (mean 32.7 ng/m³) and
have rich anthropogenic emissions sources in the environment (Yang et al., 2023). The alkyl nitrate
yield increases as the carbon chain lengths of the precursor alkanes increase (Lim et al., 2009;
Matsunaga et al., 2009; Yeh et al., 2014). The *n*-alkyl nitrate (monofunctional organic nitrate) stability
increases and the saturated vapour pressure decreases as the carbon chain length increases. Long-chain
alkyl nitrates therefore tend more than short-chain alkyl nitrates to be associated with airborne particles
and to be involved in particulate matter formation (Lim et al., 2005; Yee et al., 2012). The increasing
*n*-alkyl nitrate concentrations in the particulate phase as the *n*-alkyl nitrate carbon chain length
increased needed to be investigated further by investigating the influencing factors and the mechanisms
involved in *n*-alkyl nitrate formation.
**3.1.2 Diurnal and seasonal variations in *n*-alkyl nitrate concentrations and homologue patterns**
As shown in Table 1, the mean $C_9$–$C_{16}$ *n*-alkyl nitrate concentrations in $PM_{2.5}$ were higher at night than
in the day and the mean $C_{12}$–$C_{16}$ *n*-alkyl nitrate concentrations were significantly higher at night than in
the day (p<0.01). However, the contributions of the different *n*-alkyl nitrate homologues to the total
*n*-alkyl nitrate concentrations in the day and night samples were not significantly different, as shown in
Figures 2 and 3.
Temporal trends in the total $C_9$–$C_{16}$ *n*-alkyl nitrate concentrations during the sampling period are shown
in Figure 4. The *n*-alkyl nitrate concentrations varied seasonally, with the maximum total concentration
occurring in winter and the mean concentration decreasing in the order winter > spring > autumn >
summer. According to the analysis of variation, the contributions of the different *n*-alkyl nitrate
homologues varied seasonally, with the contributions in summer being significantly different from the
contributions in the other seasons (p<0.01) but the compositions in winter, spring, and autumn not
being significantly different. The mean particulate-bound *n*-alkyl nitrate concentrations in winter and
spring were significantly higher than the mean particulate-bound *n*-alkyl nitrate concentrations in
summer and autumn (p<0.01) based on the independent samples t-test.
We inferred that the diurnal and seasonal differences and changes in the particulate-bound *n*-alkyl
nitrate concentrations may be influenced by the meteorological factors and the changes in
particulate-bound alkyl nitrates formation process. Temperature affects the partitioning of the
semi-volatile organic compounds between the gas and particle phases, the fraction of ONs in the
particle phase increases with decreasing temperature (Kenagy et al., 2021), and the precursor *n*-alkanes
are more likely to partition into particles with the high partitioning coefficient of gas-particle
partitioning when the temperature is lower (Wick et al., 2002; Lyu et al., 2016). Other meteorological
factors such as the mixing-layer height and atmospheric dispersion conditions can also affect the
concentration level of particulate-bound alkyl nitrates by influencing the concentrations of $PM_{2.5}$ and
precursor *n*-alkanes (Wang et al., 2009; Wagner and Schäfer, 2017). However, variations in the
concentration of particulate-bound alkyl nitrates are more related to their formation (Rollins et al.,
2013). More abundant particulate matter and *n*-alkanes, influenced by meteorological factors, may
further provide the reaction conditions for the formation of particulate-bound alkyl nitrates.
In addition, the mean particulate-bound *n*-alkyl nitrate concentration was lowest in summer even
though the maximum short-chain ($C_1$–$C_5$) alkyl nitrate concentration in the gas phase was previously
found to occur in the summer (Simpson et al., 2006; Wang et al., 2013; Ling et al., 2016; Sun et al.,
2018). Long-chain particulate-bound *n*-alkyl nitrates ($C_9$-$C_{16}$) and gaseous short-chain alkyl nitrates
($C_1$-$C_5$) in the same season such as summer showed different characteristics, which may be due to their
different formation mechanisms. However, it needs to be further analyzed.
**3.2 Particulate-bound *n*-alkyl nitrate formation mechanisms**
**3.2.1 Differences between particulate-bound *n*-alkyl nitrates and gaseous alkyl nitrates**
It is generally agreed that organic nitrates are secondary products of gas-phase photochemical reactions

in the atmosphere (Perring et al., 2013; Ng et al., 2017) and that organic nitrates enter the particulate phase through gas–particle partitioning (Capouet et al., 2005; Gu et al., 2017). At high background NOx concentrations, short-chain ($C_1$–$C_5$) alkyl nitrates are mainly produced through gas-phase reactions between alkanes and OH· radicals during the day (i.e., in the presence of sunlight) (Robert, 1990; Wisthaler et al., 2008). Alkanes react with OH· radicals to form alkyl radicals through hydrogen subtraction, and the alkyl radicals are further oxidized to give $RO_2$· radicals. Finally, the $RO_2$· radicals react with nitric oxide to give alkyl nitrates. Short-chain ($C_1$–$C_5$) alkyl nitrates have been found to be secondary products of photochemical reactions, their concentrations correlate with the concentrations of photochemical pollutants and in particular to significantly positively correlate with the ozone concentration (Wang et al., 2013; Ling et al., 2016; Sun et al., 2018). Short-chain alkyl nitrate concentrations vary temporally in a similar way to the peroxyacetyl nitrate concentration, with the maximum concentration occurring in summer (Simpson et al., 2006). However, the temporal trends in particulate-bound long-chain ($C_9$-$C_{16}$) *n*-alkyl nitrate concentrations we found were different from the temporal trends in gas-phase short-chain alkyl nitrate concentrations found in previous studies.

Temporal trends in the total $C_9$–$C_{16}$ *n*-alkyl nitrate concentrations and ozone concentrations during the sampling period were compared to investigate the relationships between particulate-bound *n*-alkyl nitrates and the gas-phase reactions of photochemical process. The $C_9$–$C_{16}$ *n*-alkyl nitrate and ozone concentrations are shown in Figure 5. The total particulate-bound *n*-alkyl nitrate and ozone concentrations followed opposite temporal trends, with the lowest ozone concentration and highest total particulate-bound *n*-alkyl nitrate concentration occurring in winter, and the highest ozone concentration and lowest particulate-bound *n*-alkyl nitrate concentration occurring in summer. A significant negative correlation was found between the ozone and particulate-bound *n*-alkyl nitrate concentrations ($p<0.01$, $r=-0.411$). The $C_9$, $C_{10}$, and $C_{11}$ *n*-alkyl nitrate concentrations did not significantly correlate with the ozone concentration but the $C_{12}$–$C_{16}$ *n*-alkyl nitrate concentrations significantly negatively correlated with the ozone concentration ($p<0.01$).

$C_9$-$C_{16}$ particulate-bound *n*-alkyl nitrates showed diametrically opposite characteristics and different environmental behaviors from gaseous alkyl nitrates, which suggest that particulate-bound *n*-alkyl nitrates are not the indicators of photochemical pollution and may form through different mechanisms from gas-phase short-chain ($C_1$-$C_5$) alkyl nitrates. Research has shown that there may be other reaction pathways for the formation of particulate organic nitrates, particulate-bound organic nitrates can be formed via non-homogeneous reactions (Li et al., 2022). Therefore, we inferred that particulate-bound n-alkyl nitrates may not be formed through the gas-phase reactions in the photochemical process involving ozone and that long-chain ($C_{12}$-$C_{16}$) *n*-alkyl nitrates may not be the secondary products of gas-phase homogeneous reactions in photochemical process.

### 3.2.2 Possible particulate-bound *n*-alkyl nitrate formation mechanisms

The temporal trends in the particulate-bound *n*-alkyl nitrate and $PM_{2.5}$ concentrations are shown in Figure 4. The $C_9$–$C_{16}$ *n*-alkyl nitrate and $PM_{2.5}$ concentrations followed similar temporal trends, and the concentrations of both changed synchronously, indicating that the $C_9$–$C_{16}$ *n*-alkyl nitrate and $PM_{2.5}$ concentrations may have correlated. Statistical tests were performed, and, indeed, a significant positive correlation was found between the particulate-bound *n*-alkyl nitrate and $PM_{2.5}$ concentrations ($p<0.01$, $r=0.664$). The particulate-bound $C_9$–$C_{11}$ *n*-alkyl nitrate homologue concentrations did not significantly

correlate with the PM$_{2.5}$ concentration, and the C$_9$–C$_{11}$ $n$-alkyl nitrates and precursor $n$-alkanes were
found at low detection rates and concentrations in the PM$_{2.5}$ samples. We concluded that C$_9$–C$_{11}$ $n$-alkyl
nitrates in particulate matter may form through both gas-phase and particle-phase reactions. The
C$_{12}$–C$_{16}$ $n$-alkyl nitrate homologue concentrations significantly positively correlated with the PM$_{2.5}$
concentration ($p < 0.01$). According to previous study about particulate-bound $n$-alkanes in Beijing
(Yang et al., 2023), we found that particulate-bound $n$-alkyl nitrates showed the same temporal trends
and pollution characteristics as $n$-alkanes, the particulate-bound $n$-alkanes and PM$_{2.5}$ concentrations
significantly correlated ($p < 0.01$, $r = 0.618$). From this we hypothesize that C$_{12}$-C$_{16}$ particulate-bound
$n$-alkyl nitrate and particulate matter concentrations probably correlated because of reactions involving
precursors of $n$-alkyl nitrates on the particulate matter, meaning the particulate matter acted as a
medium on which particulate-bound $n$-alkyl nitrates formed or $n$-alkyl nitrates are involved in the
formation of particulate matter.
We found that particulate-bound $n$-alkyl nitrates may not be the products of gas-phase homogeneous
reactions, so other mechanisms may be involved in particulate-bound $n$-alkyl nitrate formation. It has
previously been found that organosulfate compounds, which have similar structures to organic nitrates,
can form through non-homogeneous reactions involving sulfate and organosulfate compound
precursors on surfaces of particles (Farmer et al., 2010). Organosulfates and organic nitrates are
important organic pollutants in particulate matter and play important roles in the formation of haze (Li
et al., 2018). Similar compounds may form through similar mechanisms, studies have shown that ONs
can be formed through non-homogeneous reactions (Zhen et al., 2022; Li et al., 2022), so we
hypothesize that particulate-bound $n$-alkyl nitrates may form through reactions between alkanes and
nitrate on particulate matter. Semi-volatile $n$-alkanes (precursors of $n$-alkyl nitrates) are widely present
in particulate matter (Kang et al., 2018; Han et al., 2018; Lyu et al., 2019; Yang et al., 2023), and the
$n$-alkane concentration in particulate matter increases as the carbon chain length increases (Aumont et
al., 2012). Abundant $n$-alkanes in particulate matter make it possible for reactions to occur to form
$n$-alkyl nitrates. Nitrogen oxides are precursors of organic nitrates and may be involved in the
formation of particulate-bound $n$-alkyl nitrates, so we compared the temporal trends in the NO$_2$ and
particulate-bound $n$-alkyl nitrate concentrations. The NO$_2$ and particulate-bound $n$-alkyl nitrate
concentrations are shown in Figure 6. The C$_9$–C$_{16}$ particulate-bound $n$-alkyl nitrate and NO$_2$
concentrations are significantly positively correlated ($p < 0.01$, $r = 0.626$). The C$_{12}$, C$_{13}$, C$_{14}$, C$_{15}$, and C$_{16}$
concentrations are significantly positively correlated with the NO$_2$ concentration ($p < 0.01$), indicating
that NO$_2$ may be involved in the formation of particulate-bound $n$-alkyl nitrates.
It has been found that the formation of nitrate (NO$_3^-$) in particulate matter is related to the presence of
NO$_2$ and that the NO$_3^-$ and NO$_2$ concentrations significantly positively correlate (Su et al., 2018). NO$_2$
in the atmosphere can be oxidized to NO$_3^-$ through non-homogeneous reactions on particulate matter
surfaces (Goodman et al., 1998), and most particulate-phase NO$_3^-$ forms through these
non-homogeneous reactions (Zhu et al., 2010). The high NO$_2$ concentrations found in the atmosphere
in urban areas mean that particulate-phase nitrate can form. Particulate-bound $n$-alkyl nitrates may
form through non-homogeneous reactions between $n$-alkanes and nitrate on particulate matter surfaces.
It has previously been found that $n$-alkanes can react with nitrate at room temperature with catalysis by
metallic copper to give alkyl nitrates (Luxenhofer et al., 1994; Luxenhofer et al., 1996). Copper is
widely present in airborne particulate matter in urban areas (Duan et al., 2014; Gonzalez et al., 2016)
and could catalyse the formation of particulate-bound $n$-alkyl nitrates. The similar temporal trends in

the particulate-bound $n$-alkyl nitrate, $n$-alkanes, PM$_{2.5}$, and NO$_2$ concentrations and the significant positive correlations between the $n$-alkyl nitrate, PM$_{2.5}$, and NO$_2$ concentrations indicate that particulate-bound $n$-alkyl nitrates may form through non-homogeneous reactions between precursor alkanes and particulate-bound nitrate on particulate matter surfaces. However, the formation mechanism needs further study.

**3.3 Contributions of particulate-bound $n$-alkyl nitrates to haze pollution**

Previous studies have shown a tight correlation between ONs content and SOA particle number concentrations, implying that ONs may play an important role in the nucleation and growth of atmospheric nanoparticles (Berkemeier et al., 2016; Yu et al., 2019). Organic nitrates have been found to contribute 2%–12% of particulate matter in SOAs (Fry et al., 2008; Rollins et al., 2012; Fry et al., 2013; Xu et al., 2015), meaning that the contributions of organic nitrates to particulate matter in the atmosphere should not be ignored and that anthropogenic precursors for long-chain particulate-bound $n$-alkyl nitrates are abundant in the environment and should therefore be of more concern than is currently the case.

The temporal trends in the particulate-bound $n$-alkyl nitrate and PM$_{2.5}$ concentrations were similar, as shown in Figure 4. The particulate-bound $n$-alkyl nitrate and PM$_{2.5}$ concentrations significantly positively correlated ($p<0.01$, $r=0.664$), indicating that particulate-bound $n$-alkyl nitrates contributed to the formation of particulate matter. The particulate-bound $n$-alkyl nitrate and PM$_{2.5}$ concentrations increased sharply during haze pollution events in winter, spring, and autumn, indicating that particulate-bound $n$-alkyl nitrates are important components of SOAs and make marked contributions to atmospheric particulate matter and haze. Similar results were found in previous studies of organic nitrates (Rollins et al., 2012). Changes in the C$_9$–C$_{16}$ particulate-bound $n$-alkyl nitrate homologue concentrations during the sampling period are shown in Figure 7. It can be seen that the temporal changes in the $n$-alkyl nitrate homologue concentrations became more similar to the temporal changes in the PM$_{2.5}$ concentration as the carbon chain length increased. Each C$_{13}$–C$_{16}$ $n$-alkyl nitrate homologue concentration significantly positively correlated with the PM$_{2.5}$ concentration ($p<0.01$), and the correlation coefficient increased as the $n$-alkyl nitrate carbon chain length increased. This indicated that the contribution of $n$-alkyl nitrates to the formation of particulate matter and haze increased as the carbon chain length increased. Because of the high background NOx concentration in ambient air in urban areas, particulate-bound $n$-alkyl nitrate SOAs can make important contributions to the particulate matter concentration and therefore to haze. The particulate-bound $n$-alkyl nitrate concentration and atmospheric visibility significantly negatively correlated ($p<0.01$, $r=-0.698$), indicating that an increase in the particulate-bound $n$-alkyl nitrate concentration will strongly decrease atmospheric visibility during a haze event. According to previous studies, organic nitrates make an important contribution to total aerosols (Xu et al., 2015) and particulate-bound ONs have a significant correlation with SOAs (Yu et al., 2019). Although it was found in our study that the mass of C$_9$-C$_{16}$ particulate-bound $n$-alkyl nitrates accounts for only a small fraction of PM$_{2.5}$ (about 1‰), they are only a small part of particulate-bound alkyl nitrates. Considering the different carbon chain lengths, carbon frame structures and functional group substitution positions, etc., as well as isomers, and the pollution characteristics and trends of C$_9$-C$_{16}$ $n$-alkyl nitrates, we believe that the effect of particulate-bound alkyl nitrates on PM$_{2.5}$ and haze formation should not be neglected. In addition, studies have shown that NOx is the key factor in the formation of atmospheric aerosols (Rollins et al., 2012), the formation of alkyl nitrates is

one of the major pathways for the conversion of NOx from radical forms into semi-permanent reservoirs (Shepson, 2007). At high NOx concentrations, the oxidation of hydrocarbon compounds in urban areas produces more than 100 different alkyl nitrates (Calvert and Madronich, 1987), Atherton and Penner calculated from model simulations that 5% of NOx can be converted to alkyl nitrates (Atherton and Penner, 1988). Therefore, we conclude that there is a strong correlation between NOx, particulate-bound alkyl nitrates and $PM_{2.5}$. Particulate-bound *n*-alkyl nitrates strongly affect haze pollution and controlling anthropogenic emissions of NOx and VOCs (precursors of particulate-bound *n*-alkyl nitrates) would effectively control particulate matter pollution and improve air quality in urban areas.

**4 Summary**

The concentrations of *n*-alkyl nitrates in $PM_{2.5}$ were determined, and all eight $C_9$–$C_{16}$ *n*-alkyl nitrate homologues were detected in $PM_{2.5}$, indicating that long-chain alkyl nitrates are present in airborne particulate matter in Beijing. The total $C_9$–$C_{16}$ *n*-alkyl nitrate concentrations during the sampling period were 9.67–2731.82 $pg/m^3$, and the mean was 578.44 $pg/m^3$. The detection rate, concentration range, and mean concentration of each *n*-alkyl nitrate homologue group in the particulate matter samples increased as the carbon chain length increased. The $C_{12}$–$C_{16}$ *n*-alkyl nitrates contributed more than the $C_9$–$C_{11}$ *n*-alkyl nitrates to the total *n*-alkyl nitrate concentrations, indicating that long-chain *n*-alkyl nitrates were more abundant than short-chain *n*-alkyl nitrates in the particulate matter. There were marked diurnal and seasonal differences in the particulate-bound *n*-alkyl nitrate concentrations. The mean $C_{12}$–$C_{16}$ *n*-alkyl nitrate concentrations were significantly higher at night than in the day ($p<0.01$). The maximum particulate-bound *n*-alkyl nitrate concentrations occurred in winter, and the mean concentrations decreased in the order winter > spring > autumn > summer. The lowest mean concentration was found in summer even though the maximum short-chain ($C_1$–$C_5$) alkyl nitrate concentrations in the gas phase have previously been found to occur in summer. The particulate-bound *n*-alkyl nitrate concentration followed the opposite temporal trend to and significantly negatively correlated with the ozone concentration. We concluded that long-chain particulate-bound *n*-alkyl nitrates may be form through different mechanisms to gas-phase short-chain alkyl nitrates and may not be the secondary products of gas-phase homogeneous reactions in photochemical process. The particulate-bound *n*-alkyl nitrate concentrations followed the same temporal trend to and significantly positively correlated with the $PM_{2.5}$ and $NO_2$ concentrations ($p<0.01$). Particulate-bound *n*-alkyl nitrates may formed through non-homogeneous reactions between alkanes and nitrate on particulate matter surfaces, meaning that particulate matter acts as a reaction substrate and reactant carrier. Particulate-bound alkyl nitrates are important contributors of airborne particulate matter and strongly affect atmospheric visibility, meaning the roles of particulate-bound alkyl nitrates in the formation of haze cannot be ignored and controlling anthropogenic emissions of precursors of particulate-bound *n*-alkyl nitrates in urban areas with high background NOx concentrations will effectively control haze pollution and improve air quality.

**Acknowledgements**

This work was supported by the National Natural Science Foundation of China [grant no. 91744206] and the Beijing Science and Technology Planning Project [Z181100005418016]. We also thank Dr.

Gareth Thomas for his help in editing this paper to improve the grammar.
**Data availability**
The data presented in this article are available from the authors upon request (junjin3799@126.com).
**Author contribution**
JJ conceived and designed the study, provided direct funding, and helped with manuscript revision.
JYY and GYL mainly conducted the sampling and sample analysis and wrote and revised the
manuscript. The other authors helped with sampling and analysis. All authors read and approved the
final manuscript.
**Competing interests**
The authors declare that they have no conflict of interest.

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

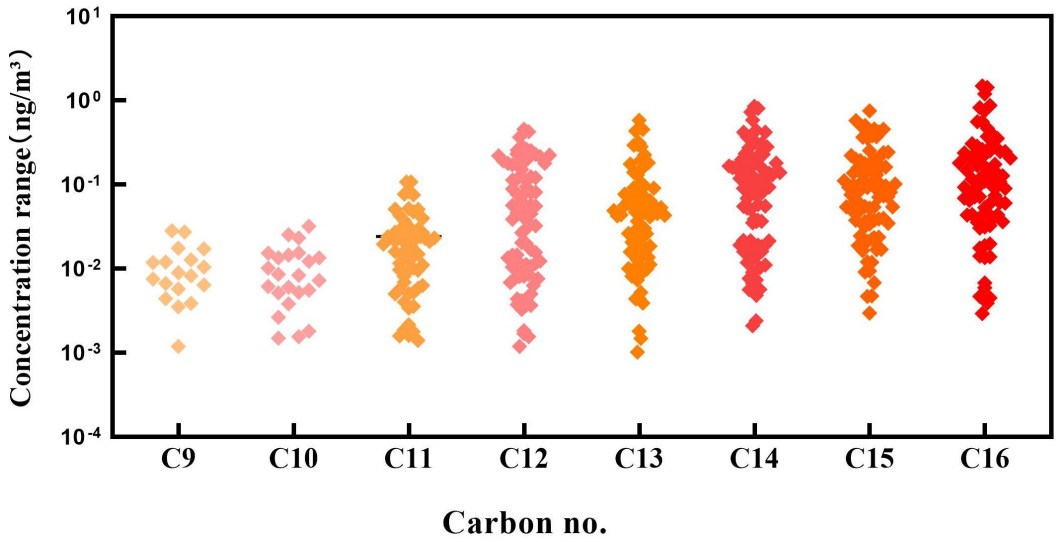


**Figure 1. Concentrations of C$_9$–C$_{16}$ *n*-alkyl nitrates in Beijing during the sampling period.**
**(The concentrations below detection limit are donated by "0")**

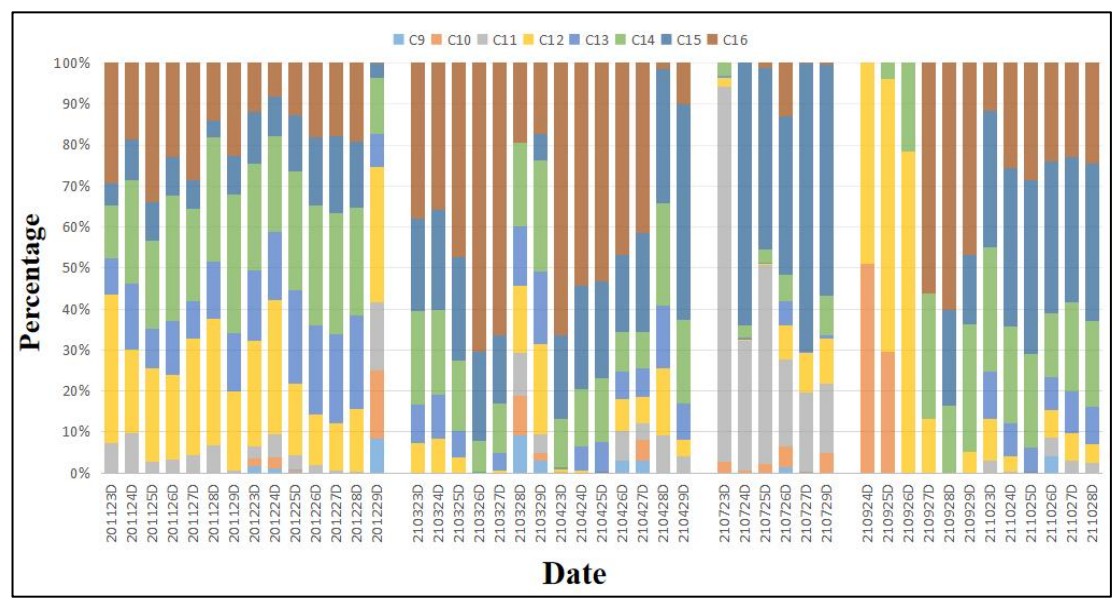


**Figure 2. Contributions of the C$_9$–C$_{16}$ *n*-alkyl nitrate homologues to the total C$_9$–C$_{16}$ *n*-alkyl nitrate**
**concentrations in the day samples collected in different seasons.**

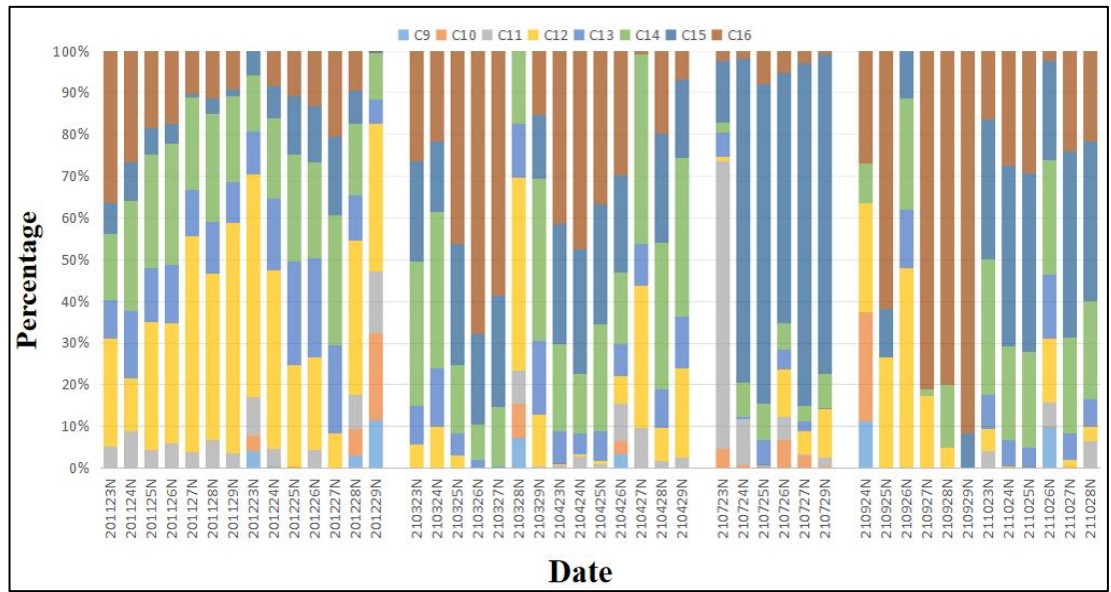

**Figure 3. Contributions of the $C_9$–$C_{16}$ *n*-alkyl nitrate homologues to the total $C_9$–$C_{16}$ *n*-alkyl nitrate**
**concentrations in the night samples collected in different seasons**

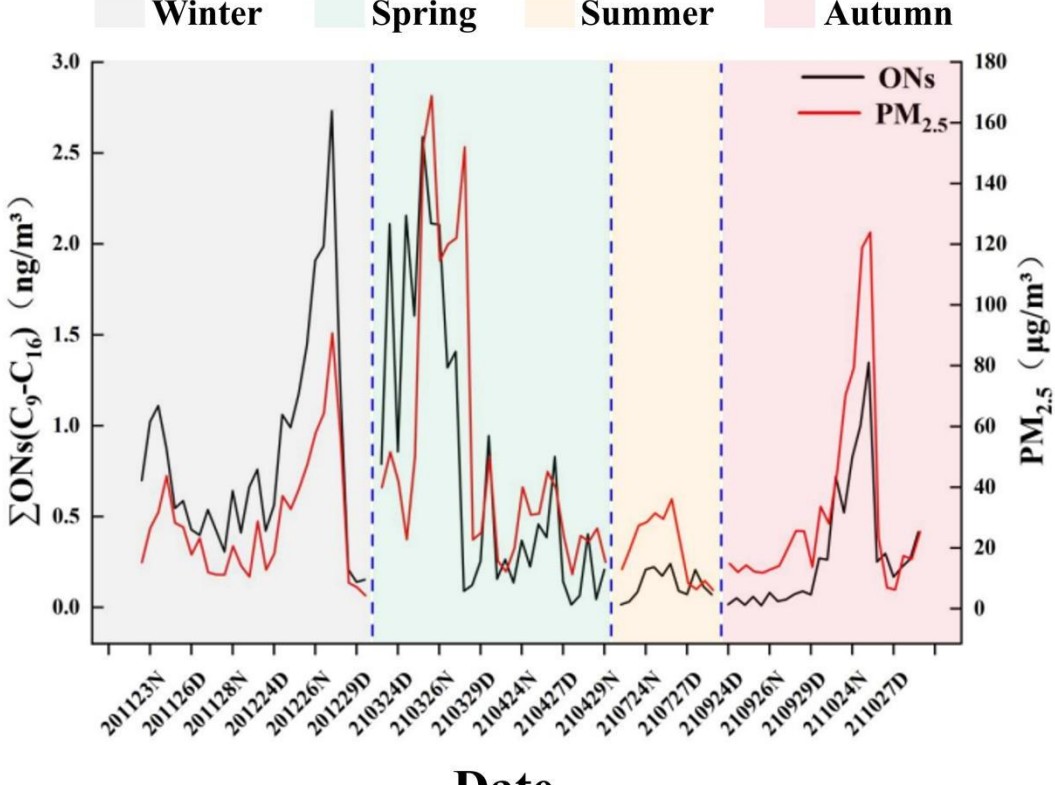

**November 23, 2020 and "N" indicates samples collected at night)**

**Figure 4. Total $C_9$–$C_{16}$ *n*-alkyl nitrate and PM2.5 concentrations during the sampling period in Beijing.**

**Figure 5. Total C₉–C₁₆ *n*-alkyl nitrate and ozone concentrations during the sampling period in Beijing.**
**(The x-axis labels are defined as the sampling time of the samples, for example, "201123" indicates date of**
**November 23, 2020; "D" indicates samples collected in the day and "N" indicates samples collected at night;**
**the dotted lines are the dividing lines and delineate the four seasons.)**

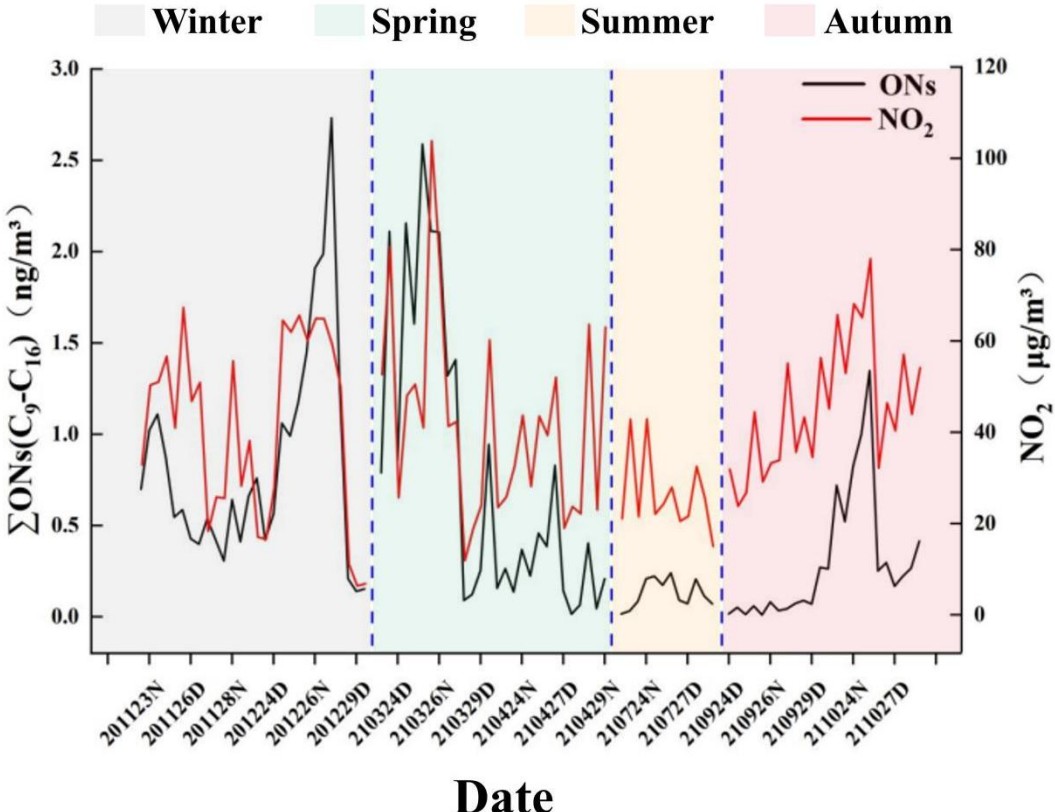

Figure 6. Total $C_9$–$C_{16}$ *n*-alkyl nitrate and $NO_2$ concentrations during the sampling period in Beijing.

(The x-axis labels are defined as the sampling time of the samples, for example, "201123" indicates date of November 23, 2020; "D" indicates samples collected in the day and "N" indicates samples collected at night; the dotted lines are the dividing lines and delineate the four seasons.)

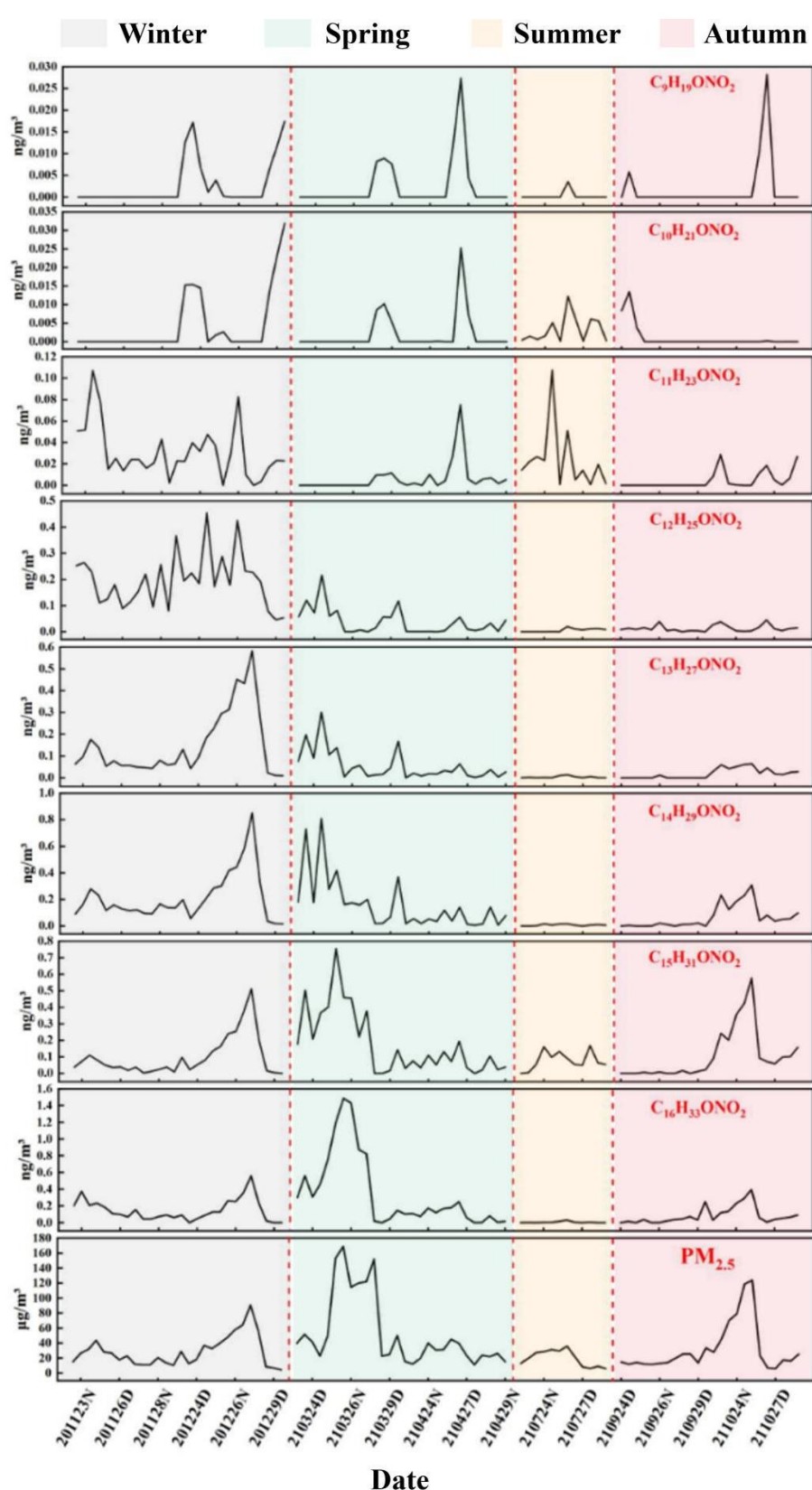

**Figure 7. C_9–C_16 *n*-alkyl nitrate homologue and PM_{2.5} concentrations during the sampling period in Beijing.**
**(The x-axis labels are defined as the sampling time of the samples, for example, "201123" indicates date of**
**November 23, 2020; "D" indicates samples collected in the day and "N" indicates samples collected at night;**
**the dotted lines are the dividing lines and delineate the four seasons.)**

 **Table 1. C$_9$–C$_{16}$ *n*-alkyl nitrate concentration ranges, mean concentrations, and detection rates**

| n-Alkyl nitrates | Concentration range(pg/m³) | | | Mean concentration(pg/m³) | | | Detection rate | | |
|---|---|---|---|---|---|---|---|---|---|
| | Day(n=46) | Night(n=46) | Total(n=92) | Day | Night | Total | Day | Night | Total |
| C$_9$H$_{19}$ONO$_2$ | ND-12.7 | ND-28.2 | ND-28.2 | 1.76 | 2.45 | 2.11 | 21.7% | 19.6% | 20.7% |
| C$_{10}$H$_{21}$ONO$_2$ | ND-23.1 | ND-32.0 | ND-32.0 | 2.44 | 2.79 | 2.61 | 34.8% | 30.4% | 32.6% |
| C$_{11}$H$_{23}$ONO$_2$ | ND-108 | ND-82.4 | ND-108 | 15.6 | 15.5 | 15.6 | 69.6% | 69.6% | 69.6% |
| C$_{12}$H$_{25}$ONO$_2$ | ND-253 | ND-454 | ND-454 | 58.8 | 91.6 | 75.2 | 93.8% | 91.3% | 92.4% |
| C$_{13}$H$_{27}$ONO$_2$ | ND-433 | ND-582 | ND-582 | 57.9 | 76.3 | 67.1 | 87.0% | 89.1% | 88.0% |
| C$_{14}$H$_{29}$ONO$_2$ | ND-586 | ND-852 | ND-852 | 104 | 160 | 132 | 95.7% | 95.7% | 95.7% |
| C$_{15}$H$_{31}$ONO$_2$ | ND-460 | ND-755 | ND-755 | 98.7 | 145 | 122 | 87.0% | 89.1% | 88.0% |
| C$_{16}$H$_{33}$ONO$_2$ | ND-1.49*10$^3$ | ND-1.43*10$^3$ | ND-1.49*10$^3$ | 156 | 190 | 173 | 89.1% | 95.7% | 92.4% |
| ∑C$_9$-C$_{16}$ | 9.67-2.11*10$^3$ | 14.6-2.73*10$^3$ | 9.67-2.73*10$^3$ | 495 | 683 | 589 | 100% | 100% | 100% |
