# Peer review of "Particulate-bound alkyl nitrate pollution and formation mechanisms in Beijing, China"

_EGUsphere, 2023_

## Author Response (AR1)

**Author's Response**

**Particulate-bound alkyl nitrate pollution and formation mechanisms in Beijing, China**

Jiyuan Yang[1*], Guoyang Lei[1*], Jinfeng Zhu[1], Yutong Wu[1], Chang Liu[1], Kai Hu[1], Junsong Bao[2], Zitong Zhang[1], Weili Lin[1] and Jun Jin[1,3].

[1]College of Life and Environmental Sciences, Minzu University of China, Beijing 100081, China

[2]State Key Laboratory of Water Environment Simulation, School of Environment, Beijing Normal University, Beijing, 100875, China

[3]Beijing Engineering Research Center of Food Environment and Public Health, Minzu University of China, Beijing 100081, China

*These authors contributed equally to this work

**Referee Comment (from Yongjie, Wei):**

This manuscript describes the n-alkyl nitrate pollution in Beijing, China. The authors detected n-alkyl nitrates from PM2.5 samples and evaluated the relationship between the formation of n-alkyl nitrate and photochemical reactions in PM2.5 by analyzing the temporal trends and correlations of n-alkyl nitrates with related pollutants, and inferred the formation mechanism of n-alkyl nitrates in particulate matter. This article presents the first qualitative and quantitative analysis of n-alkyl nitrates in PM2.5 using a new method. The manuscript is suitable for publication after the following concerns are responded.

1. The authors need to further justify the interest in studying n-alkyl nitrates in PM2.5 because it appears that the environmental impact of n-alkyl nitrates is limited.

2. Line 37-42. Please include citations in the background.

3. Line 103-105. Please provide the references of the n-alkanes emission.

4. Line 112. What is the basis for the day and night division of the sampling time?

5. Section 2.3. How to determine the quality and purity of synthetic standards of n-alkyl nitrates? Please provide more methodological details.

6. Line 181-183. How to define the long-chain and short-chain n-alkyl nitrates?

7. Line 186. Change "highest" to "largest".

8. Section 3.1.2. Insufficient explanation of the influencing factors on seasonal differences in n-alkyl nitrate concentrations in PM2.5, more discussion is needed on the impact of meteorological factors.

9. Line 246-248. For the temporal trend of particulate long-chain n-alkyl nitrate concentrations and the variability of gas-phase short-chain alkyl nitrate concentrations found in previous studies with both gas particle state and chain length variables, why are they compared together?

10. Line 267-269. The results of trend and correlation analysis of n-alkyl nitrate in PM2.5 with O3 concentration levels are not sufficient to indicate that their formation are not related to photochemical reactions.

**Author's response:**

**Dear referee:**

Thank you for your constructive comments and detailed revisions on our manuscript. We have carefully considered the suggestions and made some changes on the details of the manuscript accordingly. We supplemented the analysis in the conclusion and provided additional evidence to support the speculative ideas we presented. We have tried our best to improve this manuscript in order to it can be published successfully, please find our itemized responses and our revisions/corrections in below.

1. The authors need to further justify the interest in studying n-alkyl nitrates in PM2.5 because it appears that the environmental impact of n-alkyl nitrates is limited.

**Reply:** Thank you for your suggestions. Atmospheric particulate matter and aerosol particles directly or indirectly affect climate and regional air quality. The atmosphere is a huge oxidizing chemical reactor, and the oxidizing capacity of the atmosphere is closely related to the chemical evolution of aerosol particles (Gen et al., 2022). Alkyl nitrates in particulate matter have not received attention in the past, as a kind of semi-permanent reservoir species, the formation of alkyl nitrates consumes nitrogen oxides and thus become an important sink for atmospheric nitrogen oxides (Perring et al., 2010). In addition, alkyl nitrates may release nitrogen dioxide by photolysis, affecting the balance of nitrogen oxides (Barnes et al., 1993). The photolysis of particulate nitrate can contribute to atmospheric oxidation capacity by producing strong oxidants such as hydroxyl radicals (Gen et al., 2022). Particulate-bound alkyl nitrates as a kind of secondary pollutants affected by anthropogenic emissions have an important influence on the oxidation of the atmospheric environment and the formation of regional haze (Browne et al., 2012). Therefore, we studied the sources and pollution characteristics of particulate-bound alkyl nitrates to explore their effects on the atmospheric environment and air quality.

**Modification:** We have further explained our interest in studying particulate-bound *n*-alkyl nitrates in the manuscript (**L56-67:** As a kind of semi-permanent reservoir species, ONs are important participants in the atmospheric nitrogen cycle, which involves various atmospheric sources and sinks of nitrogen oxides. The formation of ONs consumes nitrogen oxides and atmospheric oxidants, thus becomes an important sink for atmospheric nitrogen oxides (Perring et al., 2010) and affects the atmospheric lifetimes of free radicals, the ozone concentration, and photochemical reactions (Calvert et al., 1987). In addition, ONs may release nitrogen dioxide and produce

strong oxidants such as hydroxyl radicals by photolysis, affecting the balance of nitrogen oxides in regional NOx cycles (Barnes et al., 1993; Chen et al., 1998) and contribute to atmospheric oxidation capacity (Gen et al., 2022), respectively. Semi-volatile ONs are an important kind of sources and component of secondary organic aerosols (SOAs) and contribute to fine particulate matter (PM$_{2.5}$) (Rollins et al., 2012). As important secondary air pollutants, ONs affect the oxidation in the atmosphere and the formation of haze (Browne et al., 2012), controlling particulate-bound ONs may therefore be key to controlling both PM$_{2.5}$ and ozone in the atmosphere. **L97-99:** Particulate-bound alkyl nitrates as a kind of secondary pollutants affected by anthropogenic emissions have an important influence on the oxidation of the atmospheric environment and the formation of regional haze pollution (Browne et al., 2012).).

2. Line 37-42. Please include citations in the background.

**Reply:** Thank you for pointing this out, we will add the citations.

**Modification:** We have added the citations for the background (**L38:** Li et al., 2017; **L41:** Bai et al., 2018)

3. Line 103-105. Please provide the references of the n-alkanes emission.

**Reply:** Thank you for pointing this out, we will provide the references of *n*-alkanes emission in the manuscript.

**Modification:** We have provided the references (**L114:** Kang et al., 2018; Cui et al., 2021).

4. Line 112. What is the basis for the day and night division of the sampling time?

**Reply:** Thank you for your question. We wanted to explore the differences of the concentrations of *n*-alkyl nitrates in PM$_{2.5}$ between day and night, and in order to take into account to the influences of vehicle emissions to the concentration levels of particulate-bound *n*-alkyl nitrates, the day and night sampling times are divided based on the morning and evening rush hours in Beijing.

**Modification:** We have added the timing of rush hours in Beijing in Section 2.1 (**L122-123:** according to the morning and evening rush hours in Beijing tend to be 7-9 am and 5-8 pm, respectively. ).

5. Section 2.3. How to determine the quality and purity of synthetic standards of n-alkyl nitrates? Please provide more methodological details.

**Reply:** Thank you for pointing this out. In this study, we characterized and quantified

the synthesized standards by GC-MS/MS to determine their purity. The synthesized standard and toluene were used as solute and solvent respectively to prepare the standard solution, and then detected the standard solution by GC-MS/MS full scan detection. According to the total ion flow diagram and mass spectrogram obtained by GC-MS/MS, we qualitatively analyzed the synthesized standards. In the case of *n*-nonyl nitrate, for example, as shown in the total ion flow diagram, only one compound showed high instrumental response, indicating the high purity. The characteristic ions of *n*-alkyl group and organic nitrates appeared in the mass spectrogram, and these characteristic ions all have high relative abundance. Therefore, we identified that the synthesized standard substance is *n*-nonyl nitrate. We will add more methodological details in Section 2.3.

[Figure]

Total ion flow diagram

[Figure]

Mass spectrogram

**Modification:** We have added more methodological details in this section (**L142-147:** The standards were examined and analyzed by GC-MS/MS, and detected by full scan detection. According to the total ion flow diagrams and mass spectrograms obtained by GC-MS/MS, only one compound showed high instrumental response in the total ion flow diagrams, indicating the high purity of synthesized standards. The characteristic irons of n-alkyl nitrates, $[CH_2ONO_2]^+$ ion (m/z 76.07) and $[NO_2]^+$ ion (m/z 46.07) appeared in the mass spectrograms and have high relative abundance, indicating the synthesized standards are *n*-alkyl nitrates.).

6. Line 181-183. How to define the long-chain and short-chain n-alkyl nitrates?

**Reply:** Thank you for your question. In this study, we defined $C_9$-$C_{11}$ *n*-alkyl nitrates as short-chain and $C_{12}$-$C_{16}$ *n*-alkyl nitrates as long-chain based on the carbon chain length, the detection rates in $PM_{2.5}$ samples and their pollution characteristics. We will modify the statement in this part to make it clearer.

**Modification:** We have revised the statement in this part (**L202-204:** For *n*-alkyl nitrates with a single function group, relatively long chain *n*-alkyl nitrates ($C_{12}$-$C_{16}$) are more abundant than relatively short chain *n*-alkyl nitrates ($C_9$-$C_{11}$).).

7. Line 186. Change "highest" to "largest".

**Reply:** Thank you for your suggestion, we will change the description.

**Modification:** We have changed "highest" to "largest" (**L207**).

8. Section 3.1.2. Insufficient explanation of the influencing factors on seasonal differences in n-alkyl nitrate concentrations in PM2.5, more discussion is needed on the impact of meteorological factors.

**Reply:** Thank you for pointing this out. Differences in concentrations of semi-volatile organic pollutants such as *n*-alkyl nitrate in $PM_{2.5}$ may be affected by meteorological factors such as temperature. We will add the discussion of the effect of meteorological factors on the differences in particulate-bound *n*-alkyl nitrate concentrations in Section 3.1.2.

**Modification:** We have added relevant discussion in the Section 3.1.2 (**L248-260:** We inferred that the diurnal and seasonal differences and changes in the particulate-bound *n*-alkyl nitrate concentrations may be influenced by the meteorological factors and the changes in particulate-bound alkyl nitrates formation process. Temperature affects the partitioning of the semi-volatile organic compounds between the gas and particle phases, the fraction of ONs in the particle phase increase with decreasing temperature (Kenagy et al., 2021), and the precursor *n*-alkanes are more likely to partition into particles with the high partition coefficient of gas-particle partitioning when the temperature is lower (Wick et al., 2002; Lyu et al., 2016). Other meteorological factors such as the mixing-layer height and atmospheric dispersion conditions can also affect the concentration level of particulate-bound alkyl nitrates by influencing the concentrations of $PM_{2.5}$ and precursor *n*-alkanes (Wagner and Schäfer, 2017). More abundant particulate matter and *n*-alkanes, influenced by meteorological factors, may further provide the reaction conditions for the formation of particulate-bound alkyl nitrates. **L264-266:** Long-chain particulate-bound *n*-alkyl nitrates ($C_9$-$C_{16}$) and gaseous short-chain alkyl nitrates ($C_1$-$C_5$) in the same season such as summer showed different characteristics, which maybe due to their different formation mechanisms. However, it needs to be further analyzed.).

9. Line 246-248. For the temporal trend of particulate long-chain n-alkyl nitrate concentrations and the variability of gas-phase short-chain alkyl nitrate concentrations found in previous studies with both gas particle state and chain length variables, why are they compared together?

**Reply:** Thank you for your question. We think that during the gas-particle partitioning process of alkyl nitrate, short-chain alkyl nitrates (C≤8) have higher vapor pressure and usually in the gas phase, while long-chain alkyl nitrates (C>8) can easily enter the particle phase or form directly on the particulate phase due to their lower vapor pressure. We aimed to compare the different distributions of long-chain and short-chain alkyl nitrates in the gas and particulate phases suggesting that long-chain alkyl nitrates may have a different environmental behavior than short chains.

10. Line 267-269. The results of trend and correlation analysis of n-alkyl nitrate in PM2.5 with O3 concentration levels are not sufficient to indicate that their formation are not related to photochemical reactions.

**Reply:** Thank you for pointing this out. According to previous studies of alkyl nitrates in gas phase, we found that the formation of alkyl nitrates in gas-phase environments as a branching reaction of photochemical processes (Perring et al., 2013; Sun et al., 2018), and these short-chain alkyl nitrates ($C_1$-$C_5$) showed a significant positive correlation with ozone and their peak concentration occur in summer (Wang et al., 2013; Ling et al., 2016). However, we found that long-chain *n*-alkyl nitrates in $PM_{2.5}$ showed different pollution characteristics and environmental behaviors from short-chain alkyl nitrates. Therefore, we inferred that the formation of long-chain alkyl nitrates are different from that of short-chain and may not come from the gas-phase formation reactions of photochemical process but may be formed directly in the particulate phase through the non-homogeneous reaction. This conclusion is speculative due to the limitations of experimental conditions and need further research, so we will make some modifications in the manuscript.

**Modification:** We have modified some statements in this part (**L295-303:** $C_9$-$C_{16}$ particulate-bound *n*-alkyl nitrates showed diametrically opposite characteristics and different environmental behaviors from gaseous alkyl nitrates, which suggest that particulate-bound n-alkyl nitrates are not the indicators of photochemical pollution and maybe form through different mechanisms from gas-phase short-chain ($C_1$-$C_5$) alkyl nitrates. Research have shown that there may be other reaction pathways for the formation of particulate organic nitrates, particulate-bound organic nitrates are able to form via non-homogeneous reactions (Li et al., 2022). Therefore, we inferred that particulate-bound *n*-alkyl nitrates may not be formed through the gas-phase reactions in photochemical process involving ozone and that long-chain ($C_{12}$-$C_{16}$) *n*-alkyl nitrates are not the secondary products of gas-phase homogeneous reactions in photochemical process.).

11. Section 3.2.2. The result on the formation mechanism are speculative and not sufficiently supported by evidences.

**Reply:** Thank you for pointing this out. Due to the limitations of the experimental conditions, we only studied the $C_9$-$C_{16}$ *n*-alkyl nitrates in $PM_{2.5}$ but lacked the

concentration levels of these pollutants in the gas phase. However, previous studies have not reported these *n*-alkyl nitrates ($C_9$-$C_{16}$) in the gas phase. The conclusions in this section are speculative, we will try to find more literature to support the conclusions as evidence.

**Modification:** We have revised this part and added some references as evidence to support our points (**L315-332:** According to previous study about particulate-bound *n*-alkanes in Beijing (Yang et al., 2023), we found that particulate-bound *n*-alkyl nitrates showed the same temporal trends and pollution characteristics as *n*-alkanes, the particulate-bound *n*-alkanes and $PM_{2.5}$ concentrations significantly correlated (p<0.01, r=0.618). From this we hypothesize that $C_{12}$-$C_{16}$ particulate-bound *n*-alkyl nitrate and particulate matter concentrations probably correlated because of reactions involving precursors of *n*-alkyl nitrates on the particulate matter, meaning the particulate matter acted as a medium on which particulate-bound *n*-alkyl nitrates formed or *n*-alkyl nitrates are involved in the formation of particulate matter; **L329-330:** studies have shown that ONs are able to form through non-homogeneous reactions (Zhen et al., 2022; Li et al., 2022); **L353-358:** The similar temporal trends in the particulate-bound *n*-alkyl nitrate, *n*-alkanes, $PM_{2.5}$, and $NO_2$ concentrations and the significant positive correlations between the *n*-alkyl nitrate, $PM_{2.5}$, and $NO_2$ concentrations indicate that particulate-bound *n*-alkyl nitrates maybe form through non-homogeneous reactions between precursor alkanes and particulate-bound nitrate on particulate matter surfaces. However, the formation mechanisms still needs to be further study.).

12. Line 583-585. Change "Table" to "Figure".

**Reply:** Thank you for pointing this out, we will correct this mistake.

**Modification:** We have corrected the mistake and changed "Table" to "Figure".

13. Figure 4-8. What is the definition of the dotted line, please explain.

**Reply:** Thank you for your question. The dotted lines are the dividing lines and delineate the four seasons, from left to right, winter, spring, summer, and fall. We will add the definition of the dotted line in the figures.

**Modification:** We have added the relevant definition (**L675**; **L680**; **L685**; **L690**).

14. Figure 2-8. What is the definition of horizontal coordinate numbering, please explain or modify.

**Reply:** Thank you for your question. The horizontal coordinate number in Figure 2-8 indicate the sampling time of the samples. We will add the explanation of the definition of horizontal coordinate number in the figures.

**Modification:** We have added the relevant explanation in the figures (**L664-665**; **L669-670**; **L673-674**; **L678-679**; **L683-684**; **L688-689**).

We would like to thank you again for taking the time to review our manuscript.

**Reference:**

[revised manuscript text omitted]

**Referee Comment (from Anonymous Referee):**

This is a paper describing measurement of long-chain alkyl nitrates in particulate matter in China. The measurements seem well done and I think there is probably interesting material here, however, as presented, the analysis is not mature enough to make compelling conclusions and it's hard to tell, from what is presented, whether it is merely an issue of presentation or whether additional correlative measurements would be needed. I include specific comments below but it seems like it should be possible to do more extensive calculations of the expected distributions of these

compounds between the gas and particle phases which might explain the observed diurnal and seasonal trends. Until those calculations (based on a wealth of previously published work regarding gas-particle partitioning of VOC) I don't belive that the main conclusion (formation of organic nitrates in the particle phase) can be supported. Smaller comments include:

1. Line 47 – I don't think it's true that more oxidation occurs as NOx concentrations increase. Probably under most conditions? But under NOx saturated conditions that's not true anymore. I recommend rephrasing.

2. End of the same paragraph – the last sentence doesn't really wrap these things together the way I think you want to. Maybe try to make the point that particulate-bound organic nitrates reflect both ozone production and SOA formation chemistry?

3. I recommend re-writing paragraph that starts on line 55. ONs are not mostly esters and they're not such a reactive nitrogen oxide, they're more like semi-permanent reservoir species. They're an interesting component of PM but only a large fraction under very special circumstances. If the main idea of this paragraph is to describe how ONs behave as part of the coupled Hox-NOx cycles that lead to ozone production and then how they can partition to aerosol then this should be rearranged and refocused.

4. Then, beginning of the next paragraph, in this case you mean nitrates including nitrate salts and organic nitrates? I would rephrase because, as written, that is going to be confusing to some readers since you've been only talking about organic nitrates. And then later be specific about the fraction of particulate nitrate that are observed to be ONs. Finally, the fraction of PM2.5 that's SOA seems a bit disconnected from the rest of this discussion since ONs are a minor component of both SOA and PM.

5. And then in the discussion of the different VOC that can contribute to ONs, I think it's odd not to mention the relative abundances of those precursors. Do you have measurements of any of the hydrocarbon precursors?

6. In 2.1 it might be good to mention the timing of rush hour as it relates to your sample times.

7. Section 2.3 – Have you checked the quantitative yields of the alkyl nitrate that you're using to make your standards? I think you should say more about how sure you are that they can be used as calibration standards.

8. Why is NO2+ used for confirmation and CH2ONO2 quantitatively? Can you comment on the relative detection efficiencies of the quantitative fragment for different ONs? Or I guess that is taken care of with the calibration but it's probably worth mentioning why you choose to use these ions the way you do.

9. Line 168 – 63% recovery isn't great. Was it systematically low for certain compounds? Which ones? Any idea why? The small standard deviation makes it seem like it must have been systematic.

10. Line 188 – more longer chain compounds doesn't mean that they are more abundant overall, it might just be that they're more likely to partition. Are the vapor pressures known or estimated? Can you evaluate the implied overall concentration (gas+particle) based on your observations in the particle phase using known vapor pressures and the calculated partition coefficients?

11. Lines 193 and after – Are there number-of-carbon effects to vapor pressure? I wonder if part of this trend is because c9-c11 includes only one even C number while 12-16 includes only two odds? See for example this reference: https://pubs.acs.org/doi/abs/10.1021/es0201810

12. Lines 210-213 – has anyone measured a speciated VOC distribution here? Can you see if your contributions to particulate phase match up? (I mentioned this earlier too) I'm not sure we need to call for more investigation unless we have evidence that the known chemistry fails to explain what you're seeing.

13. 3.1.2 – How much of this can be explained by colder temperatures at night?

14. Paragraph that starts on line 223 – Please be a bit more specific on which statistical tests have been used to determine significance for different things.

15. Line 234 – different chain lengths of nitrates don't have different formation mechanisms. But particulate nitrate formation is really temperature sensitive so it makes total sense that overall nitrate production might be highest in the summer even though particulate nitrate peaks in the winter. Or perhaps the winter has different VOC sources? Is this an area with isoprene emissions? Because those are very volatile products so that could drive a big summer to winter difference.

16. Figure 1 – what is the x-axis? Are the different points individual samples smushed into the same little region? Is it a function of time?

17. Figure 4 – You don't need both Figure 4 and 6. I recommend just keeping figure 6. Also why put day and night together? Maybe make them different traces? But the mass fraction of ONs are insanely small. Does current figure 6 imply that it's ~0.001% of PM? If so I think it's hard to argue, as you do at the end of the paper, that these n-alkyl nitrates are an imporatant contributor to PM and haze formation.

18. Also figures 4, 5 and 6 – would help to label or color-code with seasons since it's hard to look at numerical dates, especially since there are gaps.

**Author's response:**

**Dear referee:**

Thank you for your constructive comments on our manuscript. We have carefully considered the suggestions and revised the manuscript accordingly. We have added to the analysis of our conclusion based on your suggestions and provided more additional evidence to support our points. However, we could not add more additional measurements especially of alkyl nitrates in the gas phase due to the experimental constraints, so it is difficult to calculate the expected distributions of these compounds between the gas and particle phases without being able to obtain their concentration levels in the gas phase. We have tried our best to improve this manuscript, please find our itemized responses and revisions/corrections in below.

1. Line 47 – I don't think it's true that more oxidation occurs as NOx concentrations increase. Probably under most conditions? But under NOx saturated conditions that's not true anymore. I recommend rephrasing.

**Reply:** Thank you for your suggestions, we previously thought that atmospheric

oxidizability may increase as NOx concentrations in the atmosphere increase, with enhanced photochemical reactions and increased ozone production, did not take into account the situation that NOx reaches saturation. We will rephrase this sentence.

**Modification:** We have rephrased this sentence and changed it to "Before the NOx reach saturation, more oxidation may occurs in the atmosphere as NOx concentrations increase" (**L47-48**).

2. End of the same paragraph – the last sentence doesn't really wrap these things together the way I think you want to. Maybe try to make the point that particulate-bound organic nitrates reflect both ozone production and SOA formation chemistry?

**Reply:** Thank you for pointing this out. We will add a summary at the end of this paragraph and improve this part according to your suggestion.

**Modification:** We have added a concluding sentence at the end of this paragraph (**L54-55:** they reflect both photochemical processes of ozone production and SOA formation.).

3. I recommend re-writing paragraph that starts on line 55. ONs are not mostly esters and they're not such a reactive nitrogen oxide, they're more like semi-permanent reservoir species. They're an interesting component of PM but only a large fraction under very special circumstances. If the main idea of this paragraph is to describe how ONs behave as part of the coupled Hox-NOx cycles that lead to ozone production and then how they can partition to aerosol then this should be rearranged and refocused.

**Reply:** Thank you for your suggestions. We will rewrite this paragraph according to your advice and emphasize the impacts of particulate-bound organic nitrates as a kind of semi-permanent reservoir species on the nitrogen cycle and the atmosphere.

**Modification:** We have rewritten this paragraph (**L56-67:** As a kind of semi-permanent reservoir species, ONs are important participants in the atmospheric nitrogen cycle, which involves various atmospheric sources and sinks of nitrogen oxides. The formation of ONs consumes nitrogen oxides and atmospheric oxidants, thus becomes an important sink for atmospheric nitrogen oxides (Perring et al., 2010) and affects the atmospheric lifetimes of free radicals, the ozone concentration, and photochemical reactions (Calvert et al., 1987). In addition, ONs may release nitrogen dioxide and produce strong oxidants such as hydroxyl radicals by photolysis, affecting the balance of nitrogen oxides in regional NOx cycles (Barnes et al., 1993; Chen et al., 1998) and contribute to atmospheric oxidation capacity (Gen et al., 2022), respectively. Semi-volatile ONs are an important kind of sources and component of secondary organic aerosols (SOAs) and contribute to fine particulate matter (PM$_{2.5}$) (Rollins et al., 2012). As important secondary air pollutants, ONs affect the oxidation in the atmosphere and the formation of haze (Browne et al., 2012), controlling

particulate-bound ONs may therefore be key to controlling both $PM_{2.5}$ and ozone in the atmosphere.).

4. Then, beginning of the next paragraph, in this case you mean nitrates including nitrate salts and organic nitrates? I would rephrase because, as written, that is going to be confusing to some readers since you've been only talking about organic nitrates. And then later be specific about the fraction of particulate nitrate that are observed to be ONs. Finally, the fraction of PM2.5 that's SOA seems a bit disconnected from the rest of this discussion since ONs are a minor component of both SOA and PM.

**Reply:** Thank you for pointing this out, the "nitrates" in this part means organic nitrates, we will revise this paragraph to make it more logical.

**Modification:** We have modified this paragraph (**L70-74:** The formation of particulate-bound ONs associated with non-homogeneous reactions (Zhen et al., 2022; Li et al., 2022), especially at night was highly correlated with nitrogen oxide levels. During strong air pollution events, SOAs can contribute up to 30%-77% of $PM_{2.5}$, with particulate organic nitrates accounting for 5%-40% of the organic matter (Rollins et al., 2012; Xu et al., 2015; Sun et al., 2012). **L76-77:** The strong correlation between ONs and SOAs and the diurnal trend of ONs particle size distribution indicate the key role of particulate-bound ONs (Yu et al., 2019). )

5. And then in the discussion of the different VOC that can contribute to ONs, I think it's odd not to mention the relative abundances of those precursors. Do you have measurements of any of the hydrocarbon precursors?

**Reply:** Thank you for pointing this out. According to previous research, we found that alkanes, as the precursors of alkyl nitrates, have been found to be the most abundant species and contributing 54.1-64.7% of the total VOC concentration (Li et al., 2020),. We will add the measurement results of *n*-alkanes to this part of the manuscript.

**Modification:** We have added the the measurement results of alkanes to this part (**L83-84:** Alkanes, as the precursors of alkyl nitrates, have been found to be the most abundant species and contributing 54.1-64.7% of the total VOC concentration (Li et al., 2020)).

6. In 2.1 it might be good to mention the timing of rush hour as it relates to your sample times.

**Reply:** Thank you for your suggestion. The morning and evening rush hours in Beijing tend to be 7-9 am and 5-8 pm, respectively. We will add this to Section 2.1.

**Modification:** We have added the timing of rush hours in Beijing in Section 2.1 (**L122-123**).

7. Section 2.3 – Have you checked the quantitative yields of the alkyl nitrate that you're using to make your standards? I think you should say more about how sure you are that they can be used as calibration standards.

**Reply:** Thank you for your question and suggestion. In this study, we characterized and quantified the synthesized standards by GC-MS/MS to determine their purity. The synthesized standards were detected by the GC-MS/MS full scan detection. We qualitatively analyzed the synthesized standards, according to the total ion flow diagram and mass spectrogram obtained by GC-MS/MS. In the case of n-nonyl nitrate, for example, as shown in the total ion flow diagram, only one compound showed high instrumental response, indicating the high purity. The characteristic ions of n-alkyl group and organic nitrate appeared in the mass spectrogram, and these characteristic ions all have high relative abundance. Therefore, we identified that the synthesized standard substance is n-nonyl nitrate. We will add more methodological details in Section 2.3.

[Figure]

Total ion flow diagram

[Figure]

Mass spectrogram

**Modification:** We have added more methodological details in this section (**L142-147:** The standards were examined and analyzed by GC-MS/MS, and detected by full scan detection. According to the total ion flow diagrams and mass spectrograms obtained by GC-MS/MS, only one compound showed high instrumental response in the total ion flow diagrams, indicating the high purity of synthesized standards. The characteristic irons of $n$-alkyl nitrates, $[CH_2ONO_2]^+$ ion (m/z 76.07) and $[NO_2]^+$ ion (m/z 46.07) appeared in the mass spectrograms and have high relative abundance, indicating the synthesized standards are $n$-alkyl nitrates.).

8. Why is NO2+ used for confirmation and CH2ONO2 quantitatively? Can you comment on the relative detection efficiencies of the quantitative fragment for

different ONs? Or I guess that is taken care of with the calibration but it's probably worth mentioning why you choose to use these ions the way you do.

**Reply:** Thank you for pointing this out. In this study, we used GC-MS/MS for the detection of alkyl nitrates and chose both $[CH_2ONO_2]^+$ and $[NO_2]^+$ as qualitative and quantitative ions, where $[CH_2ONO_2]^+$ was used as the parent ion and $[NO_2]^+$ as the daughter ion. In mass spectrometry, organic nitrates formed from saturated aliphatic hydrocarbons were able to cleave to produce $[CH_2ONO_2]^+$ ion, and $[CH_2ONO_2]^+$ ion were able to further cleave to produce $[NO_2]^+$ ion, both ions have high abundance. According to previous studies, the $[NO_2]^+$ ion is widely used in the detection of alkyl nitrates as a characteristic ion of alkyl nitrates (Luxenhofer et al., 1994; Luxenhofer et al., 1996). The use of these two ions as qualitative and quantitative ions for the detection of alkyl nitrates has been studied and proven (Yang et al., 2019).

9. Line 168 – 63% recovery isn't great. Was it systematically low for certain compounds? Which ones? Any idea why? The small standard deviation makes it seem like it must have been systematic.

**Reply:** Thank you for your questions. In blank spiked recovery experiments, we found that the recovery of $C_9$-$C_{11}$ *n*-alkyl nitrates was lower than that of other *n*-alkyl nitrates, probably due to the effect of higher volatility. We will improve the pretreatment method to improve the recovery in the follow-up study.

10. Line 188 – more longer chain compounds doesn't mean that they are more abundant overall, it might just be that they're more likely to partition. Are the vapor pressures known or estimated? Can you evaluate the implied overall concentration (gas+particle) based on your observations in the particle phase using known vapor pressures and the calculated partition coefficients?

**Reply:** Thank you for pointing this out. In this study, we found that $C_{12}$-$C_{16}$ *n*-alkyl nitrates with longer carbon chains have higher detection rates and concentrations than $C_9$-$C_{11}$ *n*-alkyl nitrates in PM$_{2.5}$. Therefore, we thought that long-chain *n*-alkyl nitrates are more abundant in particulate matter, this is the conclusion based on the observations. Through our investigations, we have found that the vapor pressure of each homologue of $C_9$-$C_{16}$ *n*-alkyl nitrates is approximately $0.0 \pm 0.7$ mmHg at 25℃, and decreases with the growth of the carbon chain length. However, due to the limitations of the experimental conditions, we were unable to measure the concentration levels of $C_9$-$C_{16}$ *n*-alkyl nitrates in the gas-phase environment, so we were unable to calculate the relevant partition coefficients for further study. The total concentration and partitioning of alkyl nitrate in both phases needs to be explored by more research.

**Modification:** We have revised the statement in this part (**L202-204:** For *n*-alkyl nitrates with a single function group, relatively long chain *n*-alkyl nitrates ($C_{12}$-$C_{16}$)

are more abundant than relatively short chain n-alkyl nitrates ($C_9$-$C_{11}$).).

11. Lines 193 and after – Are there number-of-carbon effects to vapor pressure? I wonder if part of this trend is because c9-c11 includes only one even C number while 12-16 includes only two odds? See for example this reference: https://pubs.acs.org/doi/abs/10.1021/es0201810

**Reply:** Thank you for your questions. We found that the vapor pressure of $C_9$-$C_{16}$ *n*-alkyl nitrates varies only with increasing carbon chain length, independent of the number of odd or even carbons.

12. Lines 210-213 – has anyone measured a speciated VOC distribution here? Can you see if your contributions to particulate phase match up? (I mentioned this earlier too) I'm not sure we need to call for more investigation unless we have evidence that the known chemistry fails to explain what you're seeing.

**Reply:** Thank you for pointing this out. Previously, no studies have focused on the partitioning and distribution of $C_9$-$C_{16}$ *n*-alkyl nitrates between gas and particle phases. Research on alkyl nitrates has mainly focused on $C_1$-$C_5$ alkyl nitrates in the gas phase (Ling et al., 2016; Sun et al., 2018), and longer *n*-alkyl nitrates have not received enough attention and research. Therefore, we did not find any direct validation of our conclusions by the results of studies on the air particle distribution of long-chain alkyl nitrates.

13. 3.1.2 – How much of this can be explained by colder temperatures at night?

**Reply:** Thank you for your questions. Temperature influence the partition of semivolatile organic compounds between the gas and particle phases. As semivolatile organic compounds, ONs enter the particle phase more easily when the temperature is lower, the fraction of ONs in the particle phase increase with decreasing temperature (Kenagy et al., 2021). However, the extent of the direct effect of temperature is hard to be quantified. According to previous study, variations in the concentration of particulate-bound alkyl nitrates are more related to their formation (Rollins et al., 2013). We prefer to think that temperature affects the formation of alkyl nitrates on PM by influencing the partitioning of precursor *n*-alkanes between the gas and particle phases.

14. Paragraph that starts on line 223 – Please be a bit more specific on which statistical tests have been used to determine significance for different things.

**Reply:** Thank you for your suggestion. The differences between samples were characterized by independent samples t-test, paired samples t-test and one-way ANOVA. We'll add to the manuscript accordingly

**Modification:** We have added the details in the manuscript (**L242:** According to the analysis of variation; **L247:** based on the independent samples t-test.).

**Reply:** Thank you for pointing this out. $C_1$-$C_5$ alkyl nitrates have been shown to be formed from daytime gas-phase reactions of alkanes and radicals during photochemical processes (Perring et al., 2013; Sun et al., 2018). However, it has been demonstrated that organic nitrates can be formed by non-homogeneous reactions occurring on the surface of particulate matter (Li et al., 2022), $NO_2$ can form the particulate phase $NO_3^-$ via a non-homogeneous reaction (Goodman et al., 1998), and *n*-alkanes can be reacted with nitric acid to form alkyl nitrates catalyzed by copper at room temperature (Luxenhofer et al., 1994; Luxenhofer et al., 1996). In addition, it has been found that non-homogeneous reactions affect the partitioning of alkyl nitrates between the gas and particle phases, the concentration of particulate-bound alkyl nitrates increases with the organic aerosol. Since the direct relationship between particulate-bound alkyl nitrates and temperature could not be observed, the differences in the concentration of particulate-bound alkyl nitrates are mainly due to chemical factors and related to their formation (Rollins et al., 2013). Therefore, we made the speculation that particulate-bound *n*-alkyl nitrates may be formed by non-homogeneous reactions. We were unable to directly quantify the extent of the effect of temperature on particulate-bound *n*-alkyl nitrates, But the seasonal differences and trends in the concentrations of particulate-bound were the same as those of the precursor *n*-alkanes, so we thought that the seasonal differences in particulate-bound *n*-alkyl nitrates come more from precursor-related formation than the temperature effects.

**Reply:** Thank you for your question, The x-axis of Figure 1 shows the structural expression of each homologue of $C_9$-$C_{16}$ *n*-alkyl nitrates, from left to right. Figure 1 represents the concentrations range of each homologue of $C_9$-$C_{16}$ *n*-alkyl nitrate in all samples, not as a function of time. The concentrations of each homologue in the samples is represented by different colors and concentrated in the corresponding area, and we hope to represent the abundance of each homologue in the $PM_{2.5}$ in this way.

**Modification:** We have modified the x-axis label of Figure 1 and added the explanation to make it can be better understood (**L658-660**).

17.  Figure 4 – You don't need both Figure 4 and 6. I recommend just keeping figure 6. Also why put day and night together? Maybe make them different traces? But the mass fraction of ONs are insanely small. Does current figure 6 imply that it's ~0.001% of PM? If so I think it's hard to argue, as you do at the end of the paper, that these n-alkyl nitrates are an imporatant contributor to PM and haze formation.

**Reply:** Thank you for your suggestions. We would like to show the time trend of $n$-alkyl nitrates concentrations in Figure 4 and the comparison of $n$-alkyl nitrates and $PM_{2.5}$ in Figure 6. In fact, Figure 4 and Figure 6 are duplicated in the function, so we will modify according to your suggestion and only retain Figure 6. We put the day and night samples together in order to ensure the temporal continuity. $C_9$-$C_{16}$ $n$-alkyl nitrates are only a part of particulate-bound alkyl nitrates, considering the different carbon chain lengths, carbon frame structures and functional group substitution positions, etc., as well as isomers, we believe that the effect of particulate-bound alkyl nitrate on PM and haze formation should not be neglected.

**Modification:** We have modified the figures that only kept Figure 6 and adjusted the order of the figures, we also revised the corresponding part of the manuscript.

18.  Also figures 4, 5 and 6 – would help to label or color-code with seasons since it's hard to look at numerical dates, especially since there are gaps.

**Reply:** Thank you for your suggestions, we will modify these figures to better distinguish the different seasons.

**Modification:** We have modified the figures (**L671**; **L676**; **L681**; **L686**).

We would like to thank you again for taking the time to review our manuscript.


**Reply:** Thank you for your question. For alkyl nitrates, this kind of air pollutants have extensively studied that $C_1$-$C_5$ alkyl nitrates are formed from photochemical reactions,

they showed a positive correlated with ozone and have the same temporal trend (Wang et al., 2013; Ling et al., 2016; Sun et al., 2018). However, we found diametrically opposite characteristics for $C_9$-$C_{16}$ $n$-alkyl nitrate with longer carbon chains in particulate matter. Combined with the fact that particulate-bound $C_9$-$C_{16}$ $n$-alkyl nitrates exhibit diurnal differences from $C_1$-$C_5$ alkyl nitrates, we hypothesize that particulate-bound $n$-alkyl nitrates may not be formed from gas-phase reactions via photochemical process. The temporal trend of particulate-bound $n$-alkyl nitrates can indeed be influenced by many factors, so the exact mechanism of formation needs to be proven by more studies. We will modify the discussion and conclusion in this part to make it more reasonable.

**Modification:** We have revised the discussion and conclusion in this part (**L295-303:** $C_9$-$C_{16}$ particulate-bound $n$-alkyl nitrates showed diametrically opposite characteristics and different environmental behaviors from gaseous alkyl nitrates, which suggest that particulate-bound $n$-alkyl nitrates are not the indicators of photochemical pollution and maybe form through different mechanisms from gas-phase short-chain ($C_1$-$C_5$) alkyl nitrates. Research have shown that there may be other reaction pathways for the formation of particulate organic nitrates, particulate-bound organic nitrates are able to form via non-homogeneous reactions (Li et al., 2022). Therefore, we inferred that particulate-bound $n$-alkyl nitrates may not be formed through the gas-phase reactions in photochemical process involving ozone and that long-chain ($C_{12}$-$C_{16}$) $n$-alkyl nitrates are not the secondary products of gas-phase homogeneous reactions in photochemical process.).

9. 3.2.2 What about formation mechanisms from NO3 radical chemistry at night?

**Reply:** Thank you for your question. In fact, the formation mechanisms from $NO_3$ radical chemistry at night to form organic nitrates are participate mainly by olefins and aromatic hydrocarbons but do not form $n$-alkyl nitrates (Perring et al., 2013). Therefore, we did not consider this formation pathway.

10. Line 552: Yang, J. et al. appeared twice here.

**Reply:** Thank you for pointing this out, we will correct it.

**Modification:** We have removed the duplicate references.

11. Figure. 1. Are measurements below DL also present in figure.1 or only measurements above DL is present? If so, please also note the ND measurements (for example, using open markers). For me, most of the markers in the plots read zero. As your DL is 0.1~1 pg/m3 and your concentration range is 0.009~2.7 ng/m3, please consider having the concentrations plotted in the log-scale.

**Reply:** Thank you for pointing this out, the measurements below the detection limit

are donated by "0" in the Figure 1, we think that it is more intuitive to represent the abundance of $C_9$-$C_{16}$ *n*-alkyl nitrates in this way.

**Modification:** We have modified the x-axis label of Figure 1 and added the explanation to make it can be better understood (**L658-660**).

12. Figures.2~8. Make the labels on the x-axis more readable or plot them against sample dates.

**Reply:** Thank you for your suggestion, we will modify the labels on these x-axis.

**Modification:** We have modified the labels on x-axis in the figures.

We would like to thank you again for taking the time to review our manuscript.

---

## Author Response (AR2)

**Author's Response**

**Particulate-bound alkyl nitrate pollution and formation mechanisms in Beijing, China**

Jiyuan Yang[1*], Guoyang Lei[1*], Jinfeng Zhu[1], Yutong Wu[1], Chang Liu[1], Kai Hu[1], Junsong Bao[2], Zitong Zhang[1], Weili Lin[1] and Jun Jin[1,3].

[1]College of Life and Environmental Sciences, Minzu University of China, Beijing 100081, China
[2]State Key Laboratory of Water Environment Simulation, School of Environment, Beijing Normal University, Beijing, 100875, China
[3]Beijing Engineering Research Center of Food Environment and Public Health, Minzu University of China, Beijing 100081, China
*These authors contributed equally to this work

**Editor Comment:**

1.) Please comment on the fact that the alkyl nitrates range in the 1 ng/m3 range while the total aerosol accounts for >100 ug/m3. Please state clearly that these compounds do only account for 0.001% of the total PM. Please discuss the relevance of this minute contribution to PM as well as this very small contribution with respect to the statement in line 73 that ON typically make up 5 to 43% of the organic matter and the statement of a "key role of particulate-bound ONs" (line 79).

2.) Please reconsider item 11) of referee #3. Plotting the concentrations in log-scale would be very interesting if the DL is indeed 1 pg/m3.

Technical comments:
l. 48: change to "more oxidation potentially occurs"
l. 123: change to "in Beijing, which tend to be..."
l. 143 and other occasions: change "mass spectorgraphs" to "mass spectra"
l. 145 change "irons" to "ions"
l. 171 change "Method and spiked..." to "Measured and spiked ..."
l. 203 "single functional group"
l. 231 "the influencing factors and the mechanisms..."
l. 253 "high partitioning coefficient"
l. 297 "and may form..."
l. 298 "Research has shown...
l. 306 Please check numbering of Figures in the text: PM2.5 is shown in Fig 4 not Fig 6
l. 331 "we hypothesize"
l. 339 check numbering: NO2 is shown in Fig 6.
l. 348 "mean that particulate-phase..."
l. 356 "may form"

**Author's response:**

**Dear editor:**

Thank you for your constructive comments and detailed revisions on our manuscript. We have carefully considered the suggestions and made some changes on the details of the manuscript accordingly. We have tried our best to improve this manuscript in order to it can be published successfully, please find our itemized responses and our revisions/corrections in below.

1.) Please comment on the fact that the alkyl nitrates range in the 1 ng/m3 range while the total aerosol accounts for >100 ug/m3. Please state clearly that these compounds do only account for 0.001% of the total PM. Please discuss the relevance of this minute contribution to PM as well as this very small contribution with respect to the statement in line 73 that ON typically make up 5 to 43% of the organic matter and the statement of a "key role of particulate-bound ONs" (line 79).

**Reply:** Thank you for your suggestions. We will revise the manuscript accordingly based on your comments. Although it was found in our study that the mass of $C_9$-$C_{16}$ particulate-bound $n$-alkyl nitrates only account for 1‰ of $PM_{2.5}$, however $C_9$-$C_{16}$ $n$-alkyl nitrates are only a little part of particulate-bound alkyl nitrates. Base on the pollution characteristics and trends of particulate-bound alkyl nitrates represented by $C_9$-$C_{16}$ $n$-alkyl nitrates in this study, considering that there is a strong correlation between NOx, particulate-bound alkyl nitrates and $PM_{2.5}$, there are so many other particulate-bound alkyl nitrates that are unstudied and need further study. We think the effect of particulate-bound alkyl nitrates on $PM_{2.5}$ and haze formation should not be neglected and this research can provide a reference about the pollution characteristics of particulate-bound organic nitrates formed from anthropogenic emission sources and their contribution to particulate matter.

**Modification:** We have revised the manuscript accordingly based on your comments in Section 3.3 (**L388-402**: According to previous studies, organic nitrates make an important contribution to total aerosols (Xu et al., 2015) and particulate-bound ONs have a significant correlation with SOAs (Yu et al., 2019). Although it was found in our study that the mass of $C_9$-$C_{16}$ particulate-bound $n$-alkyl nitrates accounts for only a small fraction of $PM_{2.5}$ (about 1‰), they are only a small part of particulate-bound alkyl nitrates. Considering the different carbon chain lengths, carbon frame structures and functional group substitution positions, etc., as well as isomers, and the pollution

characteristics and trends of $C_9$-$C_{16}$ *n*-alkyl nitrates, we believe that the effect of particulate-bound alkyl nitrates on $PM_{2.5}$ and haze formation should not be neglected. In addition, studies have shown that NOx is the key factor in the formation of atmospheric aerosols (Rollins et al., 2012), the formation of alkyl nitrates is one of the major pathways for the conversion of NOx from radical forms into semi-permanent reservoirs (Shepson, 2007). At high NOx concentrations, the oxidation of hydrocarbon compounds in urban areas produces more than 100 different alkyl nitrates (Calvert and Madronich, 1987), Atherton and Penner calculated from model simulations that 5% of NOx can be converted to alkyl nitrates (Atherton and Penner, 1988). Therefore, we conclude that there is a strong correlation between NOx, particulate-bound alkyl nitrates and $PM_{2.5}$.).

2.) Please reconsider item 11) of referee #3. Plotting the concentrations in log-scale would be very interesting if the DL is indeed 1 pg/m3.

**Reply:** Thank you for your suggestions. In order to reflect the magnitude differences between the concentrations of the samples, we will plot the concentrations in log-scale in Figure 4 according to your comment.

**Modification:** We have modified the Figure 4 (**L678**).

[Figure]

Technical comments:
l. 48: change to "more oxidation potentially occurs"
l. 123: change to "in Beijing, which tend to be..."
l. 143 and other occasions: change "mass spectorgraphs" to "mass spectra"
l. 145 change "irons" to "ions"
l. 171 change "Method and spiked..." to "Measured and spiked ..."
l. 203 "single functional group"
l. 231 "the influencing factors and the mechanisms..."
l. 253 "high partitioning coefficient"

l. 297 "and may form..."
l. 298 "Research has shown...
l. 306 Please check numbering of Figures in the text: PM2.5 is shown in Fig 4 not Fig 6
l. 331 "we hypothesize"
l. 339 check numbering: NO2 is shown in Fig 6.
l. 348 "mean that particulate-phase..."
l. 356 "may form"
the formation
l. 357-358: "However, the formation mechanism needs further study."
l. 369 check numbering of Figure.
l. 376 check numbering of Figure.
l. 391 "we conclude ..."
l. 394 this is a Summary not Conclusions
Table 1: does it make sense to show the data with two significant digits?

**Reply:** Thank you for your comments, we will modify our manuscript according to your suggestions in the appropriate places of the article and revise the data in Table 1 from retain two digits after the decimal point to three significant digits.

**Modification:** We have modified the manuscript base on technical comments (**L48:** more oxidation potentially occurs; **L124:** in Beijing, which tend to be; **L144&L147:** mass spectra; **L146:** ions; **L172:** Measured and spiked; **L204:** single functional group; **L232:** the influencing factors and the mechanisms; **L254:** high partitioning coefficient; **L298:** and may form; **L299:** Research has shown; **L307:** Figure 4; **L331-332:** we hypothesize; **L340:** Figure 6; **L349:** mean that particulate-phase nitrate; **L357:** may form; **L358-359:** However, the formation mechanism needs further study; **L370:** Figure 4; **L377:** Figure 7; **L401:** we conclude; **L406:** Summary) and revised the Table 1 (**L711**).

We would like to thank you again for taking the time to review our manuscript.

**Reference:**

Atherton, C. S. and Penner, J. E.: The transformation of nitrogen oxides in the polluted troposphere, Tellus B, 40, 380, doi: 10.3402/tellusb.v40i5.16003, 1988.

Calvert, J. G., and Madronich, S.: Theoretical study of the initial products of the atmospheric oxidation of hydrocarbons, J. Geophys. Res. Atmos., 92, 2211-2220, doi: 10.1029/JD092iD02p02211, 1987.

Shepson, P. B.: Organic nitrates, Volatile Org. Compd. Atmos., 269-291, doi: 10.1002/9780470988657.ch7, 2007.